# Up-Regulation of Interleukin-10 in Splenic Immune Response Induced by Serotype A *Pasteurella*
*multocida*

**DOI:** 10.3390/genes13091586

**Published:** 2022-09-03

**Authors:** Haoyang Li, Meirong He, Yiwen Cheng, Junming Jiang, Weijie Yang, Zhenxing Zhang, Qi An, Si Chen, Churiga Man, Li Du, Fengyang Wang, Qiaoling Chen

**Affiliations:** Hainan Key Laboratory of Tropical Animal Reproduction & Breeding and Epidemic Disease Research, Animal Genetic Engineering Key Lab of Haikou, College of Animal Science and Technology, Hainan University, Haikou 570228, China

**Keywords:** *Pasteurella multocida*, interleukin-10, immunohistochemistry, RNA-sequencing, immune response

## Abstract

*Pasteurella multocida* (*P. multocida*) is an opportunistic pathogen that is common in livestock and poultry and leads to massive economic losses in the animal husbandry sector. In this study, we challenged mice with *P. multocida* strain HN02 by intraperitoneal injection and collected spleens to measure bacterial loads. We also performed histopathological analysis by hematoxylin and eosin (H&E) staining. Then we used RNA-sequencing (RNA-seq) to detect the mRNA expression levels in the mouse spleen and quantitative real-time PCR (qRT-PCR) to verify the sequencing data. Finally, we examined the effect of HN02 on anti-inflammatory cytokine interleukin-10 (IL-10) protein expression in the spleen through immunohistochemical analysis. The results showed that compared to those in the control group, the mouse spleens in the challenge group had lesions, and the average bacteria loads was (3.07 ± 1.09) × 10^6^ CFU (colony-forming unit)/g. The RNA-seq results determined 3653 differentially expressed genes (DEGs), and the qRT-PCR analysis revealed immune-related genes consistent with the expression trend in the sequencing data. The number and area of IL-10 positive cells substantially increased to resist inflammation in the challenge group. In conclusion, we analyzed the spleens of mice infected with *P. multocida* from multiple perspectives, and our findings lay a foundation for subsequent studies on the mechanism of pathogen-host interactions.

## 1. Introduction

*P. multocida* is a Gram-negative bacteria that infects various livestock and wild animals, mainly causing swine atrophic rhinitis, fowl cholera, and hemorrhagic septicemia, resulting in enormous economic losses worldwide [1]. In some human cases, being bitten by infected animals can lead to cellulitis or even death [2]. *P. multocida* can be classified into five capsule serotypes based on the specificity of capsule antigens (A, B, D, E, F) [3]. In China, goats are mainly infected by capsule serotypes A and B, and some reports exist that goats are infected by capsule serotypes D in Malaysia [4]. *P. multocida* contains many virulence factors, such as capsules, lipopolysaccharide (LPS), outer membrane proteins (OMPs), adhesin, etc. These virulence factors participate in the processes of bacterial growth, colonization, host infection, and cell damage. As the main components of the bacterium, the capsule and LPS play important roles in the pathogenesis of *P. multocida* [5]. For example, the virulence of *P. multocida* capsule mutants in mice was weak [5]; in the UDP-galactose (galE) mutant of *P. multocida**,* LPS biosynthesis was thoroughly weakened in mice [6]; the virulence of different serotypes of *P. multocida* capsule mutants substantially decreased in mice [6]. As an important aspect of the adhesion of *P. multocida*, the specific antigens of OmpH blocked the adhesion between *P. multocida* and bovine duct mucosa cells in vitro [7]. Current studies on the virulence factors of *P. multocida* are increasingly maturing, but further study on the host immune response is still needed.

The immune response of *P. multocida* infected hosts has been widely studied. The pulmonary response of animals infected with *P. multocida* mainly includes the expression of some inflammatory mediators and apoptosis. For instance, injection of *P. multocida* into mice induced the mRNA expression of proinflammatory cytokines, such as tumor necrosis factor α (*T**nf-**α*), interleukin 6 (*I**l-6*), *I**l-1*, *I**l-8*, and *I**l-12*, and increased the number of neutrophils [8]. *P. multocida* can further activate NLRP3 inflammasome and promote *I**l-1**β* secretion in cases of high- and low-virulence of *P. multocida* in cattle [9]. The injection of natural immune regulatory factor in mice can stimulate the release of cytokines, and the serum IL-10 levels increase rapidly [10].

IL-10 is an anti-inflammatory cytokine primarily produced by leukocytes, including T cells, B cells, monocytes, macrophages, and dendritic cells (DCs) [11,12,13]. As an important immunomodulatory factor, IL-10 plays a crucial role in the negative regulation of the immune response. In a study on the *Acinetobacter baumannii* infection mouse model, compared to that with wild-type (wt) mice, the mortality and pathological damage of *I**l-10**^−^**^/^**^−^* mice increased, and the lungs produced excess pro-inflammatory cytokines and chemokines [14]. In a study using the mouse chronic gastritis model induced by *Helicobacter pylori,* the results showed that *I**l-10* inhibited the occurrence of gastric cancer induced by *H. pylori* infection combined with alcohol intake [15]. In another study, IL-10 levels in the serum were significantly elevated at 21 and 28 days after infection, and IL-10 production was significantly elevated in spleen cells at 21, 28, 35, and 42 days after infection in a *Brucella* infection rat model. These results suggested that the cytokines secreted by Th2 cells, such as IL-10, play a key role in the immune response to chronic infection [16]. Additionally, an increase in IL-10 was also detected in a foot-and-mouth virus infection mouse model [17]. In conclusion, IL-10 is an important immunomodulator during microbial infection; however, evidence indicates IL-10 can also promote pathogen persistence and limit the immune response [14]. The detailed mechanism of how IL-10 regulates the host immune response is still not completely understood.

The main difference between Gram-negative and -positive bacteria is their membrane structure [18], which is one of the reasons for the different pathogeneses and symptoms of the host during bacterial infection. For example, as a Gram-positive bacterium, *Staphylococcus aureus* is widespread and can cause severe diseases including bacteremia, pneumonia, and cellulitis [19,20]. As a Gram-negative bacteria, *P. multocida* can cause severe infection in a wide range of animals and humans [21]. Additionally, similarities and differences exist in the immune-related pathways involved in host interactions between Gram-positive and -negative bacteria. For example, to explore a possible mechanism through which *S. aureus* damages bovine mammary glands, bovine mammary fibroblasts were infected with *S. aureus*, and the enriched DEGs involved in cytokine-cytokine receptor interactions, the MAPK signaling pathway, and the TNF signaling pathway, including the IL-17 and Nod-like receptor signaling pathways, were upregulated [22]. In a study of mouse lungs infected with *P. multocida*, the Toll-like receptor signaling pathway, cytokine-cytokine receptor interactions, the nucellar factor kappa b (NF-κB) signaling pathway, and the Nod-like receptor signaling pathway were significantly up-regulated [23]. To summarize, although different mechanisms of bacterial infection have been explored with similarity among enriched pathways, the specific genes and regulatory mechanisms still remain poorly understood. Therefore, as an important Gram-negative bacteria, the pathogenesis of *P. multocida* was of great necessity to explore.

Researchers have mostly explored the pathogenic mechanism of *P. multocida* infection in the host lungs. Being a crucial immune organ in the body, the spleen is the main site where immune cells swallow, recognize, and deliver antigens to induce an immune reaction in vivo [24]. However, the related immune reactions, mechanisms, and related changes in gene expression that occur after *P. multocida* infection in animal organisms are unknown. RNA-seq technology has been applied to obtain new insights into the molecular mechanisms and interaction relationships between hosts and pathogens; for example, in a *P. multocida* infection goat model, Zhang et al. [25] identified the effects of *P. multocida* infection on goats and specific pathogenic mechanism of *P. multocida* using RNA-seq. Additionally, IHC was used in the molecular analysis of rabbits infected with *P. multocida* [26]. These technologies are being increasingly applied. In our study, we first established a mouse spleen model of serotype A *P. multocida* infection. Second, we investigated the immune reactions occurring in the mouse spleens in response to *P. multocida* infection by bacterial load measurement, H&E staining, IHC detection, RNA-seq, and qRT-PCR, with the aim of exploring the mRNA expression profiles of mouse spleens upon *P. multocida* infection and the possible function of IL-10 in the inflammation of the spleen, so as to provide a basis for further studies on the interaction between pathogens and hosts.

## 2. Materials and Methods

### 2.1. Bacterial Strains and Culture Conditions

*P. multocida* HN02 strain (NCBI accession number: cp037865.1) was isolated in our laboratory from the lung tissue sample of a goat, who died of pneumonia in Liaoning Province, China. The species-specific strain was identified by comparing the 16S rRNA sequencing results to the sequencing data available in GenBank database. The *P. multocida* HN02 strain glycerin bacteria were removed from −20 °C frozen storage, and 0.1 mL of glycerin bacteria were inoculated in tryptone soy broth (TSB) medium (Qingdao Haibo Biotechnology Co., Ltd., Qingdao, China) with 5% (*v*/*v*) newborn bovine serum (NBS) (Zhejiang Tianhang Biotechnology Co., Ltd., Huzhou, China) and then placed into a 220 rpm thermostatic oscillator (HZQ-X100A, Shanghai Yiheng Scientific Instrument Co., Ltd., Shanghai, China) at 37 °C for 12 h recovery. Single colonies were selected for input into the PCR system for colony PCR identification. These single colonies were further inoculated in TSB with 5% (*v*/*v*) NBS for 10 h to increase cultivation after determining that the colony was serotype A *P. multocida*. In the challenge assay, the CFU of *P. multocida* was counted and the amount of bacteria was cultured to 1.5 × 10^7^ CFU/mL in combination with the results of plate colony count in the previous study. The bacteria liquid was gathered and the bacteria was washed with the sterile phosphate buffer solution (PBS) (P1020, Beijing Solaibao Technology Co., Ltd., Beijing, China). Then, 1.5 × 10^7^ CFU/mL bacteria liquid of *P. multocida* was used for infecting the mice.

### 2.2. Experimental Animals

A total of 24 healthy, specific-pathogen-free (SPF) male Kuming mice (18–22 g) were purchased from Hainan Institute of Medicine Co., Ltd. (Haikou, China); the animal license number was SYXK (Qiong) 2020–0025. The mice were placed in independently ventilated and sterile rat cages (humidity was constant, and temperature was 24 °C) and were able to obtain water and food ad libitum. After arrival, the mice had 3 days for adaptation, and then they were used for the experiments. The mice were randomly divided into two treatments: a control group (*n* = 12) and a challenge group (*n* = 12), which were infected with *P. multocida*. In each treatment, 5 mice were used for bacterial load determination, 4 for H&E staining examination and IHC analyses, and the other 3 for RNA-seq analyses. All experimental protocols were approved by the Academic Committee of Hainan University under the ethical approval code HNUAUCC-2021-00068.

### 2.3. Pathogenic of P. multocida HN02

To determine the pathogenicity of *P. multocida* HN02 strain, the mice in were intraperitoneally injected with 0.2 mL of 1.5 × 10^7^ CFU/mL *P. multocida* HN02 suspension and 0.2 mL sterile PBS in the challenge and control groups, respectively. All the mice were monitored for 7 days until they died. When the mice showed serious clinical symptoms (such as depression, appetite loss, shortness of breath, and hair disorder), they were considered to be dying and were humanely killed. In this study, we selected the cervical dislocation method, which is the most widely used method of painless execution in mouse experiments. We immediately collected the spleens of the mice that died due to infection or by being humanely killed, which we then used for determining the bacterial loads and histopathological examination. We divided spleen samples of each group into three units: we placed one unit (challenge group: 3; control group: 3) in liquid nitrogen to extract the RNA for subsequent RNA-seq, we used one unit (challenge group: 5; control group: 5) for bacterial load determination, and we soaked one unit (challenge group: 4; control group: 4) in paraformaldehyde for H&E staining to observe pathological changes and for the IHC experiment. We collected the spleen tissues immediately after the mice died, which were fixed with 4% paraformaldehyde for 24 h, dehydrated in fractional ethanol, and embedded in paraffin. We sliced the tissues into 3 μm thick sections, which were then stained with H&E. We dehydrated some of the slices, which were sealed afterwards. Thereafter, we used these slices for microscopic examination. The other part of the slices was dewaxed to water, and then we repaired them with antigen. The slices were put in 3% H_2_O_2_ to incubate for 10 min. Then, the slices were sealed with 10% (*v*/*v*) goat serum, to which we added IL-10 primary antibody (GB11534, 1:400, Wuhan Servicebio Biotechnology Co., Ltd., Wuhan, China) and incubated overnight at 4 °C. We then added HRP-labeled Goat Anti-Rabbit IgG (GB23303, 1:200, Wuhan Servicebio Biotechnology Co., Ltd., Wuhan, China) and incubated the slices with added antibodies at room temperature for 30 min. We added chromogenic agent and stained the nuclei with hematoxylin. We dehydrated the slices, treated as explained above, and sealed for IHC examination.

### 2.4. Bacterial Colonization

The mice in the challenge group died owing to infection; we collected their whole spleens to determine their bacterial loads. From the organs, we removed the redundant connective tissue and fat, and then weighed the spleens. The tissue was put into sterile 2 mL Eppendorf (EP) tubes loading of three 3 mm grinding beads. We controlled the quality of each tube by adding appropriate amount of PBS. The EP tubes were placed into tissue-grinding machine, which were levigated for 4 min until the tissue completely homogenized; during this period, all operations were sterile. The homogeneous tissue was diluted by a 10-fold gradient of PBS a dilution of 10^−3^, 10^−4^, and 10^−5^. Afterwards, the diluent was spread on tryptone soy agar (TSA) medium (Qingdao Haibo Biotechnology Co., Ltd., Qingdao, China), to which we added 5% (*v*/*v*) serum, in three replicates, which we incubated in a 37 °C incubator for 24 h. Then, we counted these TSA mediums for detailed bacterial counts. The average values of number of *P. multocida* CFU were used for the statistical analysis.

### 2.5. Total RNA Extraction, Library Construction, and Sequencing

Total RNA of mice spleen tissue samples was extracted by Trizol kit (12183555, Thermo Fisher Technology (China) Co., Ltd., Shanghai, China) in accordance with the instructions of the manufacturer. The quality and concentration of RNA was examined by 1% (*w*/*v*) agarose gel electrophoresis and a spectrophotometer. The integrity of the RNA was tested using an Agilent 2100 Bioanalyzer (Agilent Technologies (China) Co., Ltd., Beijing, China). The RNA without ribosome was randomly interrupted and re-transcribed into first- and second-strand cDNA, in accordance with the instructions of the reverse transcription kit (KR118, Tiangen Biotech (Beijing) Co., Ltd., Beijing, China), after qualifying the RNA examination. Afterwards, the quality of the library was evaluated by qRT-PCR. The cDNA library was sequenced based on Sequencing By Synthesis (SBS) technology on an Illumina high-throughput sequencing platform. Clean reads were obtained by controlling the quality of the sequencing bases; the contents of Q20 and Q30 and the GC content of the clean reads were simultaneously calculated. The sequencing connectors and low-quality sequencing data introduced during library construction were removed to ensure that the quality of the reads was reliable. The screened sequences were mapped with those of the *M**us musculus* reference genome GRCm38 using HISAT2. The mapped reads were assembled using StringTie to build a more complete transcript and better evaluate the expression. The data from this study have been deposited to the NCBI Sequence Read Archive database with accession number PRJNA837087.

### 2.6. Differentially Expressed Genes Analysis

Fragments per kilobase of transcript per million fragments mapped (FPKM) was adopted as the indicator for measuring the level of gene expression, to ensure the number of reads truly reflected the level of transcript expression. Estimate size factors in the DEseq2 R package (version 1.6.3) were applied for DEGs analysis, while nbinom test was used to calculate the corrected *p*-value and fold change (FC). False Discovery Rate (FDR) < 0.01 and FC ≥ 1.5 were set up as the threshold value to screen DEGs. Heatmap or volcano map in the R package was used in our clustering analysis to reflect the expression differences between the treatment and control groups for every gene.

### 2.7. Gene Ontology and Kyoto Encyclopedia of Genes and Genomes Enrichment Analyses

DEGs were mapped to the Gene Ontology (GO) and Kyoto Encyclopedia of Genes and Genomes (KEGG) databases to obtain respective classifications. Considering the pathways in the KEGG database as the units, the results of pathway significance enrichment analysis showed the significantly enriched pathways in applying the hypergeometric test, by comparing to the whole genomic background. The target genes were mapped to the proteins in the database to search for homologous proteins. The interaction network was built according to the interaction relationships of the homologous proteins. The constructed completed protein interaction network could be guided into Cytoscape (version 3.7.2) for visualization.

### 2.8. Quantitative Real-Time PCR

Total RNA from mice spleens was extracted and re-transcribed to cDNA using a HiScript qRT SuperMix kit. We randomly selected 8 immune-related genes (*Il-**10*, *M**lkl*, *Sdhd*, *Irf1*, *Cxcl10*, *Cxcl13*, *Ccr9*, and *Cxcl2*) to verify the validity of the RNA-seq results. The criterion to select genes was based on the immune reaction function and meaning in the RNA-seq results. *β-actin* was chosen as the reference gene to normalize the transcription level of the target gene. The specific primer was designed using PrimerBLAST, according to the reference sequence in NCBI, and the criteria were: (a) the size of PCR production was 80–200 bp, (b) melting temperature was 60 ± 2 °C, and (c) the primer had to span exon–exon linkages. The relative expression of the genes is represented as the ratio of the target gene to the reference gene, calculated by the 2^−^^∆∆ct^ method. The primer sequences are shown in Table 1.

### 2.9. Statistical Analysis

All data are presented as mean ± standard error of the means (SEM). The nbinom test in DESeq2 was applied for the calculation of the *p*-value and FCs. The difference was considered significant when *p*-value < 0.05.

## 3. Results

### 3.1. P. multocida Infection in Mice

Five mice died within 24 h due to infection after challenge. All other mice showed depressed spirit, shaggy hair, and trembling and were insensitive to external stimulation in the challenge group (Figure 1A). Therefore, we considered these mice articulo mortis, so we sacrificed them humanely. For comparison, the mice in control group showed no symptoms (Figure 1B). We selected five mice that died in the challenge group and five mice in the control group to determine the bacterial loads. The autopsy results showed that the mouse spleens in challenge group had turned black, showing hyperemia and swelling (Figure 1C); in the control group, the mouse spleens showed no pathological changes (Figure 1D). The results of bacterial load analysis showed that the spleens of the five mice in the challenge group all had large amounts of bacterial colonization, with an average of bacteria load of (3.07 ± 1.09) × 10^6^ CFU/g (Table 2). The spleens of the mice in the control group contained no bacteria. The results of H&E staining showed that the splenic tissue capsule of the mice in the challenge group was rough (Figure 2A). We observed that many of the lymphocytes and macrophages in the red and white pulp were necrotic and denaturated in the challenge group. The cell body was swollen, the cytoplasm was vacuolar, and the nucleus was broken. The demarcation between red and white pulp was unclear. The number and volume of white pulp were reduced; the splenic artery was not obvious. We observed diffuse extramedullary hematopoietic cells in the red pulp (Figure 2B). For comparison, in the control group, we observed no pathological injury in the mouse spleens (Figure 2C,D). The results of IHC showed that compared with the control group, the number and area of positive cells were substantially higher, and the IL-10-positive signal was mostly expressed in the cytoplasm of macrophages in the challenge group (Figure 3A). While in the control group, IL-10 signaling did not remarkably change (Figure 3B).

### 3.2. Transcriptomic Sequencing Analysis

We generated a total of six samples, by three biological replicates under two treatments. A total of 158,407,896 clean reads were produced by the raw reads of the six samples after disposal, of which the control group generated 75,097,101 clean reads, and the challenge group generated 83,310,795 clean reads. The Q30 base percentage of all samples was above 93.92% (Table 3). The mapping efficiency was between 94.96% and 95.96% when we mapped the clean reads to the reference genome.

### 3.3. Identification of DEGs

We screened 3653 DEGs, and Figure 4 plots the result as a volcano map. Among them, 1795 and 1858 genes were up- and down-regulated, respectively. Then, the DEGs were used for cluster hierarchy analysis in this study. The relationship between the samples was calculated by the expression of DEGs and was displayed as a heatmap (Figure 5A). The results showed that the intragroup repeatability of the challenge and control groups were higher, and we found significant differences before and after challenge. To understand the biological function of the DEGs, GO enrichment and KEGG enrichment were applied to annotate the function of the DEGs. We identified two (viral protein interaction with cytokine and cytokine receptor and TNF signaling pathway) and five (intestinal immune network for IgA production, NF-κB signaling pathway, chemokine signaling pathway, C-type lectin receptor signaling pathway, and cytokine-cytokine receptor interaction) KEGG pathways (*p*-value < 0.05) among the up- and down-regulated DEGs, respectively (Figure 5B). In Figure 6, the GO-enrichment result showed that cellular process (1052 up-regulated DEGs; 853 down-regulated DEGs) had the highest number of DEGs in the biological process; in terms of cell components, the cell (1079 up-regulated DEGs; 910 down-regulated DEGs) had the largest number of DEGs. In terms of molecular function, the binding (973 up-regulated DEGs; 919 down-regulated DEGs) had the largest number of DEGs. Detailed information is provided in Appendix A.

The results of KEGG enrichment analysis showed that endocytosis (29 up-regulated DEGs; 22 down-regulated DEGs) was the most enriched in transport and catabolism; in the environmental information processing pathways, the pi3K-Akt signaling pathway (41 up-regulated DEGs; 30 down-regulated DEGs) was the most enriched; in the global and overview maps, carbon metabolism (43 up-regulated DEGs; 4 down-regulated DEGs) was the most enriched; in the immune system, the hematopoietic cell lineage (12 up-regulated DEGs; 33 down-regulated DEGs) was the most enriched (Figure 7). Detailed information is provided in Appendix A.

### 3.4. DEGs Related to Host Immune Response against Infection

In this study, to explore the immune response pathways used against *P. multocida* infection in the host spleen, we further screened the DEGs in the GO and KEGG databases with *p*-value < 0.05 as the screening condition. The results of GO-enrichment analysis showed that the terms related to immunization were positive regulation of T cell differentiation, response to bacterium, T cell receptor signaling pathway, positive regulation of IL-8 production, immune response, and positive regulation of I-κB kinase/NF-κB signaling. The results of KEGG enrichment analysis screened a total of 60 terms with significant differential expression (Appendix A). The first 20 terms with the smallest *p* value are displayed in Figure 8. In these 60 terms, the terms related to immunization were viral protein interaction with cytokine and cytokine receptor, TNF signaling pathway, cytokine-cytokine receptor interaction, hematopoietic cell lineage, and C-type lectin receptor signaling pathway. We filtered 142 immune-related DEGs from the immune pathways, and we drew a protein-protein interaction network (PPI) network using String and Cytoscape software (Figure 9).

### 3.5. Experimental Validation of DEGs

We randomly selected eight genes related to viral protein interactions with cytokines and cytokine receptor pathway, and TNF signaling pathway to evaluate the accuracy and reliability of the RNA-seq result. The results of the qRT-PCR validation showed that the mRNA expressions of *I**l-10*, *M**lkl*, *S**dhd*, *C**xcl2*, and *Cxcl13* were up-regulated, while those of *Cxcl10*, *C**cr9*, and *I**rf1* were down-regulated. These results are all consistent with the RNA-seq data, which confirmed the reliability of the RNA-seq data in this study (Figure 10).

## 4. Discussion

As an opportunistic pathogen, *P. multocida* creates many problems owing to its strong latent capacity, including difficulty of cleaning up during farming process, seriously threatening the public health safety, and hindering the development of animal husbandry [3]. In this study, we established a mouse model of *P. multocida* infection through intraperitoneal injection (Figure 11A), and we explored the possible host infection pathway based on RNA-seq data.

When we infected the mice with *P. multocida*, they all showed severe clinical symptoms. In the challenge group, the results of the autopsy of the dead mice showed splenic hyperemia and swelling and abundant bacteria colonies in the spleen. The above results indicated that we successfully constructed a mouse spleen model of *P. multocida* infection. In our study, the quantity and volume of the white pulp of the spleen in the challenge group were much smaller compared with those of the control group. The boundary between the red and white pulp was vague, and the sequencing data showed that *T**nfr1* was significantly up-regulated. These results are consistent with those of Lalić et al. in a LPS-stimulated *T**nfr1*^−/−^ mouse spleen model [27], indicating that using mice as an animal model to study the infection mechanism of *P. multocida* is reasonable; The spleen, as the most important peripheral immune organ of the body, was of value in our study.

The structure of the serotype A *P. multocida* capsule is similar to the glycosaminoglycan (GAG) of mammalian cells [28,29]. The capsule of *P. multocida* uses GAG for molecular camouflage, which not only protects the bacteria from being devoured but also disguises it as the host, which increases the toxicity of *P. multocida* [30]. The active component of the capsule of serotype A *P. multocida* is hyaluronic acid (HA). HA uses molecular camouflage to enable bacteria to evade immune defense systems and continuedly infect the host [31]. Researchers have quantitatively determined the uptake rate of neutrophils by polygonide was 3.8%, which was as high as 90% after removing the capsular HA with HA enzymes, indicating that the capsular HA of *P. multocida* inhibits the phagocytic activity of neutrophils [32]. In the present study, we speculated that the molecular camouflage of HA and the high similarity between the GAGs of type A *P. multocida*, and those host cells might be the mechanism through which *P. multocida* evades host immunity. The result of IHC analysis showed that the IL-10-positive signal was mostly expressed in the macrophages in mouse spleen in the challenge group. H&E result indicated that plenty of the immune cells of the spleen; the red and white pulp of the spleens were necrotic in the challenge group. In conclusion, we thought that in our infection model, the HA of the *P. multocida* capsule invaded the spleen of the host owing to immune escape, and inhibited the phagocytic activity of macrophages, neutrophils, and other immune cells, eventually leading to the partial necrosis of the spleen. The anti-inflammatory cytokine IL-10 was secreted by the lymphocytes and acted on the macrophages in the red pulp of the spleen to inhibit the secretion of pro-inflammatory cytokines.

The LPS of *P. multocida* was able to stimulate the immune response of a host [33]. Lipid A is an endotoxin component of LPS and can be recognized by Toll-like receptors (TLRs). CD14 receptors on phagocytes bind to LPS and lipopolysaccharide binding protein (LBP) to form a complex, which binds to LPS-binding protein *LY96*. The process promotes the oligomerization of *TLR-4* and activates intracellular signaling cascades and the NF-κB signaling pathway, ultimately leading to the expression of proinflammatory cytokines [34]. In addition to RNA-seq data, the mRNA expression of *L**bp*, *C**d14*, *L**y96*, and *T**lr-4* in the challenge group were significantly higher compared with those in the control group. We speculated that in this study, lipid A of *P. multocida* LPS first combined with the up-regulated LBP of the plasma protein to form LPS/LBP. As a surface antigen, CD14 can mediate immune response to the LPS of bacteria and combine with LPS/LBP to form a compound that acts in combination with up-regulated LY96 and activated TLR-4. The result of RNA-seq indicated that the MAPK, PI3K-AKT, and NF-κB signaling pathways in the challenge group were significantly different from those in the control group. In our infection model, up-regulated *T**lr-4* might have activated these pathways to eventually lead to the expressions of proinflammatory cytokiness such as CXCL1, CXCL2, etc.

Compared with the previous studies by our laboratory, we found out the mRNA expression profile in *P. multocida* infection were similar to some extent. For example, mice challenged with *P. multocida* serotype D HN01 strain, of which mRNA expression profile showed that *I**l-10* (*p* < 0.05) was significantly up-regulated in the challenge group’s mouse spleen, and results also revealed the high pertinence between *I**l-10* and the IL-1 family, suggesting the possibly antagonistic regulation of spleen inflammation might exist between them [35], and this finding was the same as the result of this research. Interestingly, compared with our result, a similar discovery was revealed in another study of goat bronchial epithelial cells stimulated by *P. multocida* serotype A and D [36], of which the result indicated that in the *P. multocida* strain serotype A-stimulated group, the significant KEGG pathways related to immunity were the GABAergic synapse and Toll-like receptor signaling pathway. While in the *P. multocida* strain serotype D-stimulated group, phagosome and B cell receptor signaling pathway were significantly enriched. In our research, we found that these four KEGG pathways all enriched in the mouse infection *P. multocida* strain serotype A model with *p* value > 0.05, indicating that the similarities and differences might exist when comparing the expression profile of a *P. multocida* strain to another, which meant research should be focused on in the future. We will further elaborate on the gene expression profile of this experiment in the following.

In this study, we identified the DEGs and pathways through RNA-seq analysis of infected mice. Our purpose in this study was to explore the immune response mechanism to host infection with *P. multocida*, so we focused on the significantly different immune-related pathways and genes. Our sequencing results showed that the most differentially expressed gene was *I**l-1**r2*. *I**l-1**r2* was first found in B lymphocytes and bone marrow monocytes [37]. IL-1R2 negatively regulates IL-1-dependent activity through different modes of action. IL-1R2 acts as a molecular trap for IL-1 to inhibit its activity by binding to IL-1α and IL-1β, with high affinity in a nonsignaling manner [38,39]. IL-1R2 is also involved in the anti-inflammatory effects of multiple mediators. For example, glucocorticoids, prostaglandins, aspirin, anti-inflammatory Th2 cytokines (IL-4 and IL-13), IL-10, and IL-27. These mediators promote the surface expression and release of IL-1R2 in bone marrow monocytes both in vitro and in vivo [38,40,41,42], and IL-1R2 has anti-inflammatory effects in vivo. Overexpression of *IL-1R2* had protective effects in rabbit arthritis or mouse heart transplantation models [43,44]. Sequencing data showed that *IL-1R2* expression was significantly up-regulated in the spleens of the mice in the challenge group compared with those in the control group, indicating that *IL-1R2* was secreted in infected mice to resist bacterial invasion (Figure 11B). Among the numerous DEGs, the expression of the interleukin family (*IL18*, *IL**-10*, *IL1A*) were significantly up-regulated, and *IL**-10* was the most up-regulated. The interleukins (ILs) are a family of cytokine proteins, which play an important role in immune regulation. ILs can exert their immunomodulatory function by binding to their receptors. ILs are also able to regulate growth, differentiation, and activation during the immune response. The expressions of *I**l-6* and *I**l-10* in the mouse spleens inoculated with LPS increased in a previous study, consistent with our results [45]. The results of sequencing also showed that the mRNA expression of *I**l-18* was significantly increased. The combination of IL-10 and IL-18 enhanced the cell proliferation and cytotoxicity of natural killer (NK) cells [46]. We think that the combination of the significantly increased the expression level of anti-inflammatory cytokine IL-10 with that of IL-18 activates the massive proliferation of NK cells, exerts a cytotoxic effect, eliminates harmful substances, and inhibits the inflammatory response of the host. Our sequencing data showed that the mRNA expression of *C**xcl1*, *C**xcl2*, *C**xcl14*, *C**cl5*, and other pro-inflammatory cytokines significantly increased, while the mRNA expression of *I**l-10**ra* was significantly down-regulated. Danger signals such as pro-inflammatory cytokines can induce dendritic (DC) cell maturation [47,48,49]. After DC cells matured, IL-10RA was down-regulated to produce higher levels of proinflammatory mediators [50]. We hypothesized that in this mouse infection model, *P. multocida* infection led to the up-regulation of a variety of proinflammatory cytokines, which stimulated DC cells to mature and down-regulated IL-10RA to secrete proinflammatory mediators. IL-10 plays an anti-inflammatory role; however, too much of the proinflammatory medium still caused inflammation in mouse spleen, which affected the whole body and ultimately led to death (Figure 11B).

The results of our KEGG analysis of the DEGs showed that the most significantly differential pathways included viral protein interaction with cytokines and cytokine receptors, the TNF signaling pathway, etc. The TNF signaling pathway mediates a variety of intracellular signaling pathways and inflammatory responses; regulates immune function by activating and recruiting immune cells; and can trigger cell proliferation, differentiation, apoptosis, and necrosis [51]. Therefore, we thought that various immune cells were secreted and activated in the mouse spleens during infection to resist inflammation and maintain the homeostasis of the internal environment. As a downstream gene of the TNF signaling pathway, *V**cam-1* is a major regulator of leukocyte adhesion and transendothelial migration [52,53]. In this study, the mRNA expression of *V**cam-1* was significantly higher after *P. multocida* infection. In another study, the LPS stimulation of chickens significantly increased the mRNA expression of *V**cam-1* in the spleen [54], which is consistent with our study, suggesting that the LPS of *P. multocida* may play a vital role in the mouse spleen. According to the results, we found a significant difference in the viral protein interaction with cytokines and cytokine receptors, among which the expressions of chemokines such as *C**cl5, C**xcl1*, and *C**xcl9* were significantly increased. Chemokines and chemokine receptors play an important role in the immune defense system by controlling the migration, activation, differentiation, and survival of leukocytes [55,56]. LPS stimulation in mice can significantly increase the expression levels of chemokines and receptors such as *C**cl5* and *C**xcl9* in the spleen [27].

## 5. Conclusions

In this study, we successfully constructed a goat serotype A *P. multocida* infection mouse model. We first found inflammatory lesions caused by *P. multocida* HN02 strain infection in mice by RNA-seq and then profiled the gene expression of mice infected with *P. multocida*. We focused on the changes in the expression levels of interleukin-family-related pathways and immune genes and speculated on the possible mechanism through which *P. multocida* infects the mouse spleen. These results indicated that the spleen infection with *P. multocida* produced a complex immune response. Our findings established a theoretical foundation for the further study of the pathogenesis of *P. multocida*.

## Figures and Tables

**Figure 1 genes-13-01586-f001:**
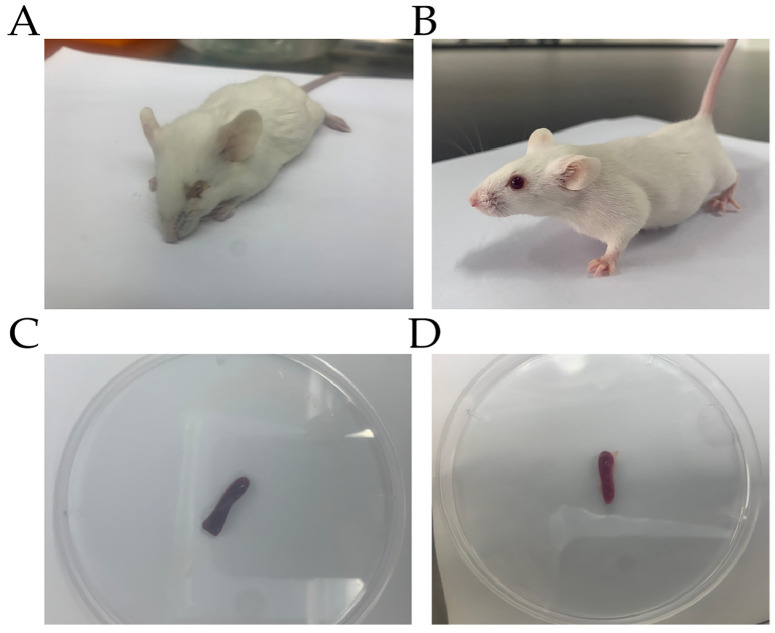
Mice and spleens, respectively, in challenge group (**A**,**C**) and control group (**B**,**D**). (**A**) Mouse showed depression and conjunctivitis. (**B**) Control group mice showed no symptoms. (**C**) Mouse spleen turned black, showing hyperemia and swelling. (**D**) Spleen in control group showed no pathological changes.

**Figure 2 genes-13-01586-f002:**
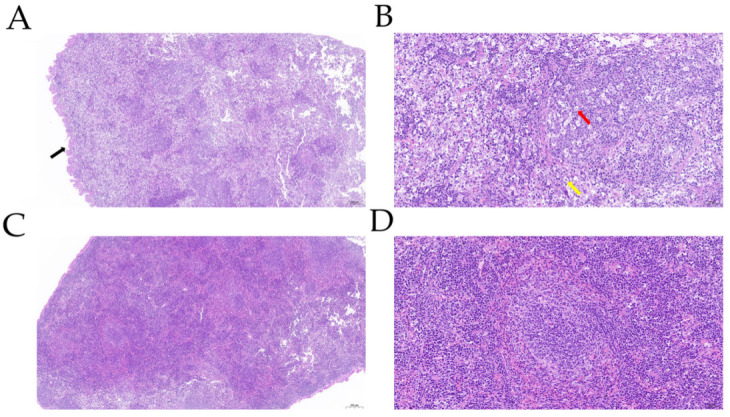
Histopathological analysis of spleen lesions in challenge group (**A**,**B**) and control group (**C**,**D**). All the tissues were stained with hematoxylin and eosin. (**A**) Splenic tissue capsule of mice in challenge group was rough (black arrow), 5× magnification. (**B**) Many lymphocytes in red and white pulp were necrotic and denatured, and nuclei were broken (red arrow). Cell body of macrophages was swollen, and cytoplasm was vacuolar (yellow arrow), 20× magnification. (**C**) The structure and morphology of spleen tissue in control group were more intact, 5× magnification. (**D**) Boundary between red and white pulp was distinct, 20× magnification.

**Figure 3 genes-13-01586-f003:**
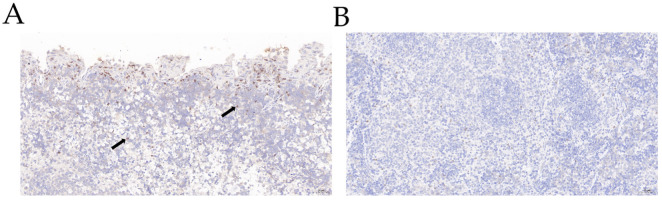
IL-10 immunodetection in challenge group (**A**) and control group (**B**). (**A**) IL-10 positive signal in mouse spleens from challenge group was increased significantly and was mostly expressed in the cytoplasm of macrophages (black arrow), 20× magnification. (**B**) No obvious aggregation of IL-10-positive cells group, 20× magnification.

**Figure 4 genes-13-01586-f004:**
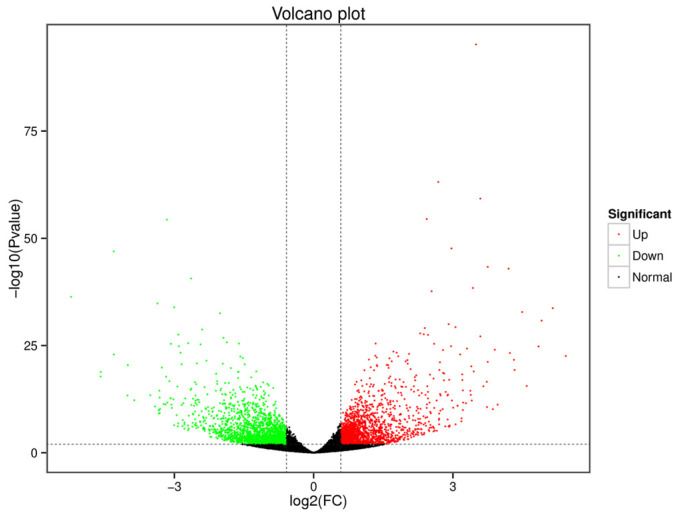
Results of DEGs. Each dot represents a gene. Green and red dots represent down- and up-regulated genes, respectively. The greater the absolute value of the abscissa was, the difference between challenge group and control group was more obvious. Similarly, the bigger the ordinate value was, the more significant the gene expression difference between the two groups.

**Figure 5 genes-13-01586-f005:**
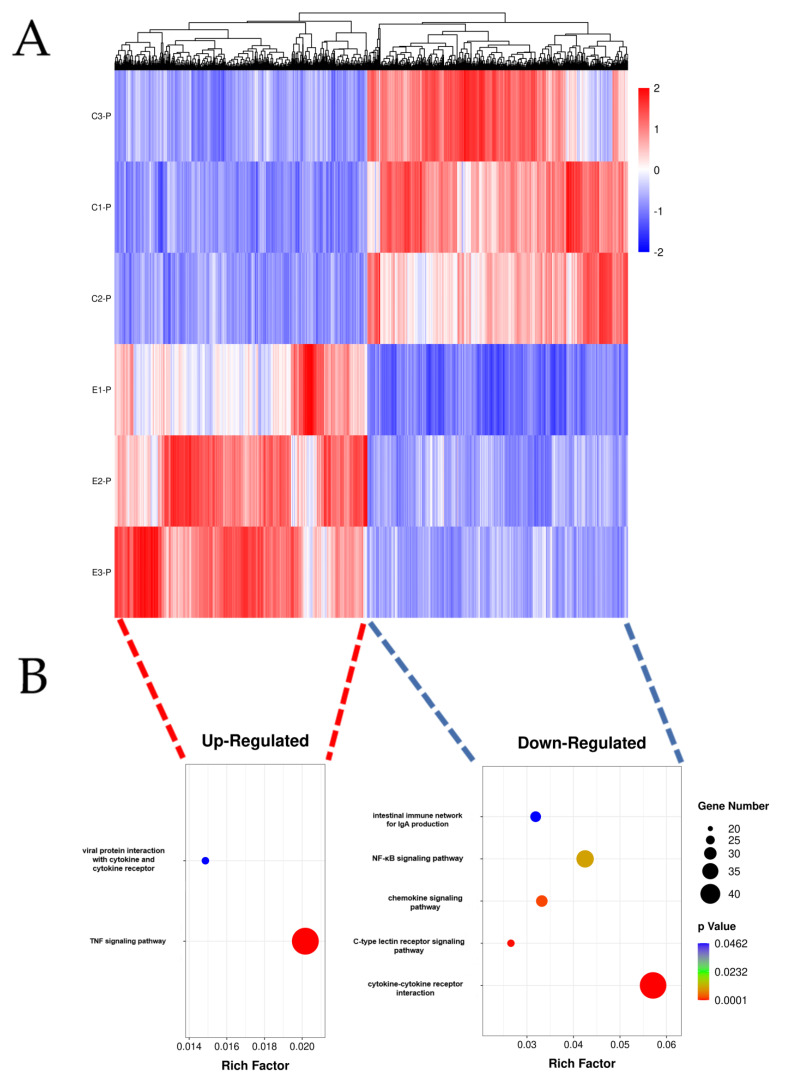
Analysis of DEGs and pathways derived from mouse spleens infected with *P. multocida*. (**A**) Heatmap of DEGs. Colors from blue (low) to red (high) represent DEG expression level. Each row in the map represents a sample, and each column represents a gene. (**B**) Kyoto Encyclopedia of Genes and Genomes (KEGG) pathway analysis of up- and down-regulated DEGs. Dot size represents the number of genes. The rich factor indicates the ratio between the number of genes annotated for a KEGG pathway and the number of all genes annotated for the KEGG pathways.

**Figure 6 genes-13-01586-f006:**
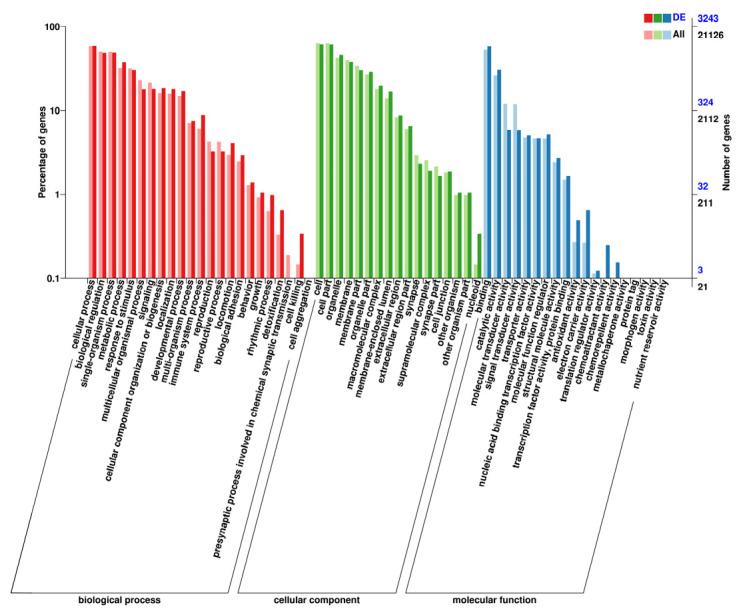
Results of Gene Ontology (GO) enrichment are classified according to different functions. DEGs and all genes are distinguished between dark and light colors, while percentages of genes are represented by histograms.

**Figure 7 genes-13-01586-f007:**
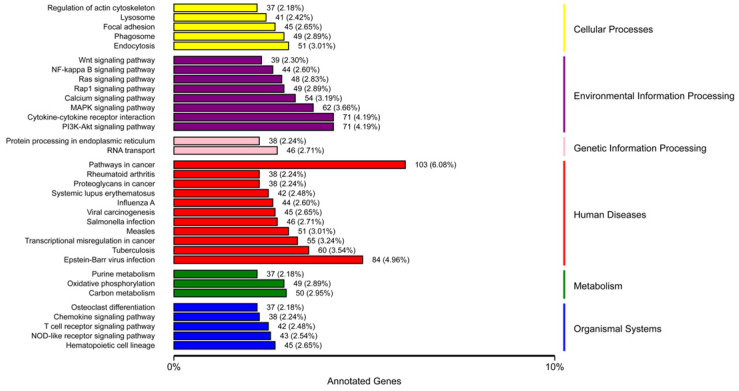
KEGG enrichment results classified according to pathway types, which are represented by different colors. Detailed quantity and proportion are shown right after histograms.

**Figure 8 genes-13-01586-f008:**
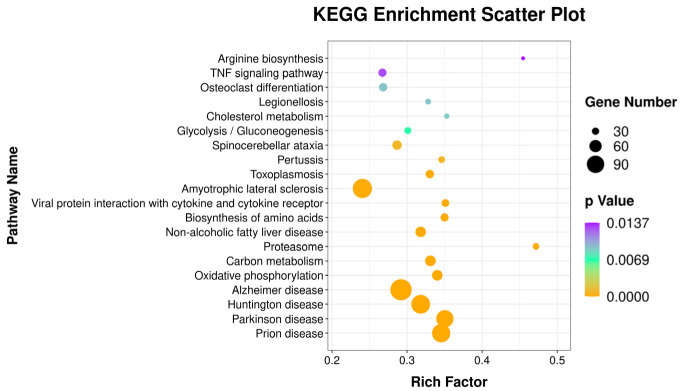
First 20 KEGG-enriched pathways, calculated numbers of DEGs in each pathway, and rich factors and *p* values. Gene numbers were represented by the size of the dots. The color of the dot went from yellow to purple, indicating the *p*-value went from small to large.

**Figure 9 genes-13-01586-f009:**
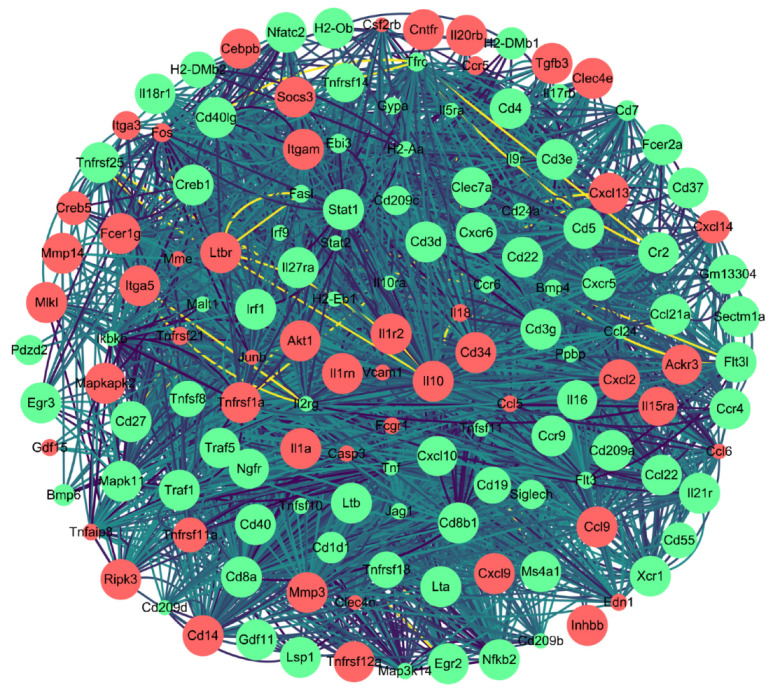
A total of 142 DEGs related to immune response were screened. Red, up-regulated genes; green, down-regulated genes. Depth of the line represents the degree of association, where a deeper line represents the closer the relationship between two proteins was, and size of the dot represents the significance of *p* values, where a larger size indicates higher significance.

**Figure 10 genes-13-01586-f010:**
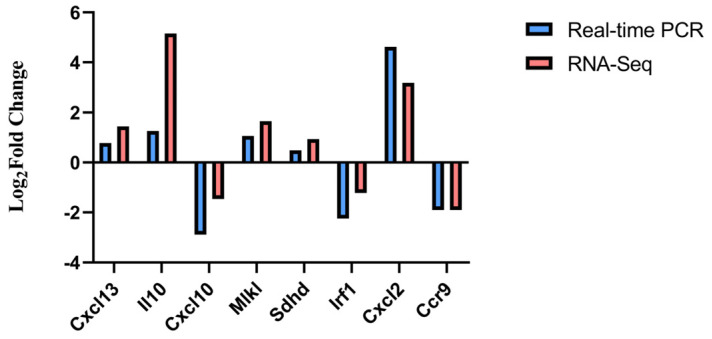
Result of comparing quantitative real-time polymerase chain reaction (qRT-PCR) with RNA-sequencing (RNA-Seq) based on the value of log_2_ fold change (FC).

**Figure 11 genes-13-01586-f011:**
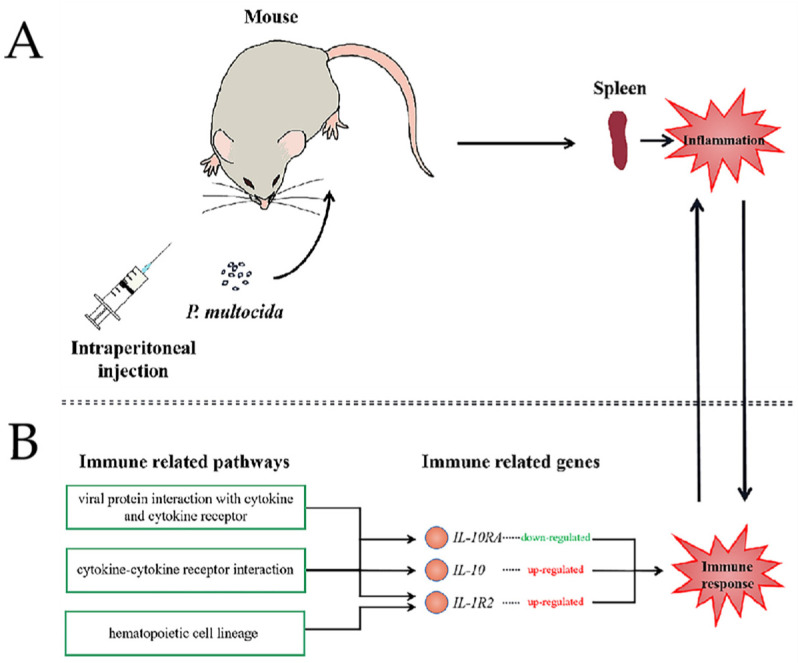
Schematic diagram of body changes caused by *P. multocida* infection in mouse spleen. (**A**) Mouse spleen model of *P. multocida* infection. (**B**) Part of the known relationship between interleukin-family-related pathways and genes.

**Table 1 genes-13-01586-t001:** Primer information.

Name of the Primers	Primer Sequences (5′-3′)	Product Length (bp)
*Il-10*-F	ACTTGGGTTGCCAAGCCTTA	161
*Il-10*-R	GAGAAATCGATGACAGCGCC	
*Mlkl*-F	AGATCCCATTTGAAGGCTGTGA	174
*Mlkl*-R	CTCATGGGCACGACACTCAT	
*Sdhd*-F	CCAAGCCACCACTCTGGTTC	134
*Sdhd*-R	GCAGCCAGAGAGTAGTCCAC	
*Irf1*-F	ACTCGAATGCGGATGAGACC	93
*Irf1*-R	CTGCTTTGTATCGGCTTTATTGA	
*Cxcl10*-F	CCACGTGTTGAGATCATTGCC	184
*Cxcl10*-R	GAGGCTCTCTGCTGTCCATC	
*Cxcl13*-F	CTCCAGGCCACGGTATTCTG	118
*Cxcl13*-R	CCAGGGGGCGTAACTTGAAT	
*Ccr9*-F	TGGAGGCTGGTCTGCATTATC	95
*Ccr9*-R	CATGCCAGGAATAAGGCTTGTG	
*Cxcl2*-F	CCCAGACAGAAGTCATAGCCAC	162
*Cxcl2*-R	TGGTTCTTCCGTTGAGGGAC	

**Table 2 genes-13-01586-t002:** Bacterial loads of mice spleens in challenge group.

Animal Number	Weight (g)	Bacterial Loading (CFU × 10^6^/g)	Average Bacterial Loading (CFU × 10^6^/g)
Challenge group-1	0.10	4.60 ± 0.46	3.07 ± 1.09
Challenge group-2	0.08	1.63 ± 0.54	
Challenge group-3	0.06	3.30 ± 0.29	
Challenge group-4	0.07	4.00 ± 1.94	
Challenge group-5	0.10	1.80 ± 0.61	

**Table 3 genes-13-01586-t003:** Results of transcriptomic sequencing.

Sample ID	Clean Reads	Clean Bases	GC (%)	Q30 (%)
C1-P	23,759,030	7,104,206,622	49.81	93.93
C2-P	27,750,220	8,310,925,154	50.62	94.24
C3-P	23,587,851	7,057,533,332	49.86	94.24
E1-P	30,059,831	9,003,329,146	50.05	94.06
E2-P	25,039,488	7,491,410,612	50.01	93.92
E3-P	28,211,476	8,439,900,430	50.51	94.18

## Data Availability

The raw data from this study have been deposited to the NCBI Sequence Read Archive database with accession number PRJNA837087.

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
