# Peer review of "Up-Regulation of Interleukin-10 in Splenic Immune Response Induced by Serotype A *Pasteurella"

_genes, 2022, doi:10.3390/genes13091586_

Round 1

Reviewer 1 Report

This manuscript was written very carefully and well organized, with almost no significant language and editing problems inside. In this paper, the authors described a typical gene functional analysis result in the P. multocida infected mouse model. After i.p. challenge with the bacteria, the spleen was taken and the total RNA was isolated for RNA seq analysis. After DEGs and pathway analysis, a bunch of inflammatory pathways showed up. The analysis methods showed no problem and were well done.

However, there are two major issues for this paper:

1.         why do we need a set of data for a mouse inflammation-to-die pathway analysis in a gram-negative bacterial infection? This was very similar to other bacteria infection results.

2.         The authors try to probe IL-10 expression in the biopsy and try to connect the pathogenesis with the IL-10 over-expression. However, did this only showed in P. multocida infection? How about the IL-10 level in other bacteria infections?

There are two minor issues that can be improved:

1.         For the GO or pathway analysis results presentation, Fig 6 and Fig 7, please describe the indicated and enriched genes or pathways that were up-regulated or down-regulated in the challenge group. 

2.         For the animal experiment, please provide a clear review ID by the Review board.

Author Response

Reviewer1

Comments and Suggestions for Authors

This manuscript was written very carefully and well organized, with almost no significant language and editing problems inside. In this paper, the authors described a typical gene functional analysis result in the P. multocida infected mouse model. After i.p. challenge with the bacteria, the spleen was taken and the total RNA was isolated for RNA seq analysis. After DEGs and pathway analysis, a bunch of inflammatory pathways showed up. The analysis methods showed no problem and were well done. 

However, there are two major issues for this paper:

  1. why do we need a set of data for a mouse inflammation-to-die pathway analysis in a gram-negative bacterial infection? This was very similar to other bacteria infection results.

Bacteria were classified into gram-negative and gram-positive bacteria based on differences in their membrane structure. Gram-negative bacteria had two membranes separated by periplasm, while gram-positive bacteria had only one membrane and one peptidoglycan [1]. In gram-negative bacteria, lipopolysaccharide (LPS) played a key role in initial adhesion and subsequent biofilm formation [2]. As an endotoxin, high concentrations of LPS might cause sepsis and septic shock [3]. While studies on the outer membrane of gram-positive bacteria have focused more on omv (outer membrane vesicles) in recent years. Omv was mainly involved in stress response, biofilm formation, immune regulation and etc [1]. In conclusion, due to the difference in the composition of the outer membrane of gram-negative bacteria and gram-positive bacteria, the pathogenesis and symptoms of the host would also be different. For example, as a gram-positive bacterium, Staphylococcus aureus was widespread and could cause severe diseases including bacteremia, pneumonia and cellulitis [10,11]; while P. multocida could cause infections in a wide range of animals and humans as a gram-negative bacteria [12]. 

Additionally, there were similarities and differences in the immune-related pathways involved in host interaction between gram-positive and gram-negative bacteria. For example, in order to explore the possible mechanism of the damage of bovine mammary gland caused by staphylococcus aureus, bovine mammary fibroblasts were infected with staphylococcus aureus, and the enriched differentially expressed genes (DEGs) involved in cytokine-cytokine receptor interaction, MAPK signaling pathway, TNF signaling pathway, among which IL-17 signaling pathway and Nod-like receptor signaling pathway were up-regulated [5]; mice lungs were infected with klebsiella pneumoniae and the results showed that IL-17 signaling pathway, TNF signaling pathway, NF-kB signaling pathway, MAPK signaling pathway were significantly enriched [6]; in a study of mice lung infected with P. multocida, Toll-like receptor signaling pathway, cytokine-cytokine receptor interaction, NF-kB signaling pathway, Nod-like receptor signaling pathway were significantly up-regulated [7]; similarly, chicken lungs were infected with avian and bovine P. multocida, Toll-like receptor signaling pathway, TNF signaling pathway, cytokine-cytokine receptor interaction were also significantly enriched [8]. To sum up, even if different mechanisms of bacteria infection were explored, some of the enriched pathways might be the same, but the specific genes and regulatory mechanisms still remained difference. As an important gram-negative bacteria that could cause zoonosis, P. multocida had high research value. Studies have shown that humans with compromised immune systems could become infected by living with bacteria-carrying animals for a long time even if they were not attacked by animals carrying P. multocida [9]. Therefore, how to control and eliminate P. multocida has gradually become one of the difficult problems facing public health. In this study, mice infected with P. multocida from sheep were used as a model, hoping to provide a theoretical basis for the prevention and treatment of P. multocida by studying the infection mechanism of mice.

1. Cao, Y.; Lin, H.C. Characterization and function of membrane vesicles in Gram-positive bacteria. Appl Microbiol Biotechnol. 2021, 105(5), 1795-1801.

2. Lee, K.J.; Lee, M.A.; Hwang, W.; Park, H.N.; Lee, K.H. Deacylated lipopolysaccharides inhibit biofilm formation by Gram-negative bacteria. Biofouling. 2016, 32(7), 711-23.

3. Hung,Y,L,; Fang, S.H.; Wang, S.C.; Cheng, W.C.; Liu, P.L.; Su, C.C.; Chen, C.S.; Huang, M.Y.; Hua, K.F.; Shen, K.H.; Wang, Y.T.; Suzuki, K.; Li, C. Corylin protects LPS-induced sepsis and attenuates LPS-induced inflammatory response. Sci Rep. 2017, 7, 46299.

4. D'Amico,F.; Casalino, G.; Bozzo, G.; Camarda, A.; Lombardi, R.; Dimuccio, M.M.; Circella, Spreading of Pasteurella multocida Infection in a Pet Rabbit Breeding and Possible Implications on Healed Bunnies. Vet Sci. 2022, 9(6), 301.

5. Miao,Z.; et al. Transcriptome sequencing reveals fibrotic associated-genes involved in bovine mammary fibroblasts with Staphylococcus aureus. Int J Biochem Cell Biol. 2020, 121, 105696.

6. Zheng, X.Y.; et al. Time-Course Transcriptome Analysis of Lungs From Mice Infected With Hypervirulent Klebsiella pneumoniae via Aerosolized Intratracheal Inoculation. Front Cell Infect Microbiol. 2022, 12, 833080.

7. Wu.C.; Qin, X.; Li, P.; Pan, T.; Ren, W.; Li, N.; Peng, Transcriptomic Analysis on Responses of Murine Lungs to Pasteurella multocida Infection. Front Cell Infect Microbiol. 2017, 7, 251.

8. Li,P.; He, F.; Wu, C.; Zhao, G.; Hardwidge, P.R.; Li, N.; Peng, Transcriptomic Analysis of Chicken Lungs Infected With Avian and Bovine Pasteurella multocida Serotype A. Front Vet Sci. 2020, 7,452.

9. Rueda, Prada.L.; Cardozo, M.; Hudson, A.; McDermott, M.; Urbina, Verjel. D.C.; Dumic, I. Disseminated Pasteurella multocida in a patient with liver cirrhosis and spontaneous bacterial peritonitis - The role of cirrhosis-associated immune dysfunction. IDCases. 2022, 29, e01542.

10. Kobayashi,S.D.; Malachowa, N.; DeLeo, F. Pathogenesis of Staphylococcus aureus abscesses. Am J Pathol. 2015, 185(6), 1518-27.

11. Lowy,F. How Staphylococcus aureus adapts to its host. N Engl J Med. 2011, 364(21), 1987-90.

12. Peng,; Wang, X.; Zhou, R.; Chen, H.; Wilson, B.A.; Wu, B. Pasteurella multocida: Genotypes and Genomics. Microbiol. Mol. Biol.Rev. 2019, 83, e00014-19.

  1. The authors try to probe IL-10 expression in the biopsy and try to connect the pathogenesis with the IL-10 over-expression. However, did this only showed in P. multocida infection? How about the IL-10 level in other bacteria infections?

In addition to the findings in this study that P. multocida could cause increased mRNA expression of IL-10 in the mouse spleen, similar findings have been found in other microbial infection models. For example, in a study of a brucella infection rat model, significantly elevated IL-10 levels were observed in the serum at 21 and 28 days after infection, and significantly increased IL-10 production in spleen cells at 21, 28, 35, and 42 days after infection, suggesting that cytokines secreted by Th2 cells such as IL-10 played an immune response to chronic infection. [13]. An increase in IL-10 was also detected in a foot-and-mouth disease virus (fmdv) infection mouse model. The overexpression of IL-10 might be associated with reduction of lymphocytes[14].

13. Khatun, M.M.; Islam, M.A.; Baek, B.K. In Vitro and In Vivo IFN-γ and IL-10 Measurement in Experimental Brucella abortus Biotype 1 Infection in Sprague-Dawley Rats. Vector Borne Zoonotic Dis. 2021, 21(8), 579-585.

14. Guo, Z.J.; Zhao, Y.; Zhang, Z.D.; Li, Y.M. Interleukin-10-Mediated Lymphopenia Caused by Acute Infection with Foot-and-Mouth Disease Virus in Mice. Viruses. 2021, 13(12), 2358.

There are two minor issues that can be improved:

  1. For the GO or pathway analysis results presentation, Fig 6 and Fig 7, please describe the indicated and enriched genes or pathways that were up-regulated or down-regulated in the challenge group.

In consideration of article length, detailed up-regulated or down-regulated genes information would be added to Supplementary Table S1 and Supplementary Table S2. We thought it would be better and more meaningful to show the most number of genes enriched GO terms and KEGG pathways in the article. The description of up-regulated or down-regulated genes in challenge group were added in detail in the results as following:

Line 253 was corrected to “In figure 6, GO enrichment result showed that, cellular process (up-regulated DEGs:1052, down-regulated DEGs:853) had the most number of DEGs in biological process; in terms of cell components, cell (up-regulated DEGs:1079, down-regulated DEGs:910) had the most number of DEGs while in molecular function, binding (up-regulated DEGs:973, down-regulated DEGs:919) had the most number of DEGs.”

Line 258 was corrected to “KEGG enrichment result showed that endocytosis (up-regulated DEGs:29, down-regulated DEGs:22) was top 1 enriched in transport and catabolism; in environmental information processing pathways, the enrichment of pi3K-Akt signaling pathway (up-regulated DEGs:41, down-regulated DEGs:30) was the most significant; in global and overview maps, the enrichment of carbon metabolism (up-regulated DEGs:43, down-regulated DEGs:4) was the most significant; in immune system, hematopoietic cell lineage (up-regulated DEGs:12, down-regulated DEGs:33) was top1 enriched (Figure 7).

  1. For the animal experiment, please provide a clear review ID by the Review board.

Line 106 was corrected to “All experimental protocols were approved by the Academic Committee of  Hainan University under the ethical approval code HNUAUCC-2021-00068.”

Reviewer 2 Report

The language used within the manuscript needs to be profoundly check& corrected, as it is influencing on the quality of understanding of the text.

please add "spleen" to the key words

please re-write the Introduction part dedicated to IL-10, to make it more clear& give more practical examples

please describe the technique of sacrifying of the aniamls. How long after natural death of animals the organs were collected?

Please add the details of Ab used for IHC.

The Reviewer would advice ELISA test for cytokins check.

Fig. 2& 3 please replace the slides with more distinct. The visual effect esp. of Fig. 3 is poor. 

Part 3.5 is too briefly explained.

line 307-308- number of animals & restricted number fo tests are not the basis to conlude about the animal model. What was the host infection pathway?

In the results part there is a lack of histological details and post-mortem observations of infected mice.

No blood/serum tests done?

How was IHC analyzed? Image -J?

Please add the slide with necrotic spleen cells.

The Authors are making lot of speculations on the basis of sequencing analysis & in silico analysis.

Why Authors are mentioning viral proteins discussing bacterial infection?

Author Response

Reviewer2 

Comments and Suggestions for Authors

The language used within the manuscript needs to be profoundly check& corrected, as it is influencing on the quality of understanding of the text.

Yes, we have checked the overall manuscript carefully and corrected several minor errors.

please add "spleen" to the key words

Yes, we have already added “spleen” to the key words.

please re-write the Introduction part dedicated to IL-10, to make it more clear& give more practical examples

Line 59 was corrected to “IL-10 is an anti-inflammatory cytokine produced primarily by leukocyte, including T cells, B cells, monocytes, macrophages and dendritic cells (DCs) [1,2,3]. As an important immunomodulatory factor, IL-10 plays a crucial part in the negative regulation of immune response. In a study of acinetobacter baumannii infection mice model, compared with wild type (wt) mice, the mortality of IL10-/- knockout mice was increased, pathological damage aggravated and the lungs produced excess pro-inflammatory cytokines and chemokines [4]. In a mouse chronic gastritis model induced by helicobacter pylori, the result showed that IL-10 inhibited the occurrence of gastric cancer induced by helicobacter pylori infection combined with alcohol intake [5]. In another study, IL-10 levels in the serum were significantly elevated at 21 and 28 days after infection, and IL-10 production was significantly elevated in spleen cells at 21, 28, 35, and 42 days after infection in a brucella infection rat model, suggesting that cytokines secreted by Th2 cells such as IL-10 played an immune response to chronic infection [6]. What’s more, an increase in IL-10 was also detected in a foot-and-mouth disease virus (fmdv) infection mouse model [7]. In conclusion, IL-10 is an important immunomodulator during microbial infection, however, there was evidence indicating IL-10 was also a double-edged sword that could both promote pathogen persistence and limit immune response [4]. The research on the detailed mechanism of IL-10 regulating host immune response is still not completely clarified.”

1. Medzhitov, R.; Shevach, E.M.; Trinchieri, G.; Mellor, A.L.; Munn, D.H.; Gordon, S.; Libby, P.; Hansson, G.K.; Shortman, K.; Dong, C.; Gabrilovich, D.; Gabryšová, L.; Howes, A.; O'Garra, A. Highlights of 10 years of immunology in Nature Reviews Immunology. Nat Rev Immunol. 2011, 11(10), 693-702.

2. Moore, K.W.; de. Waal, Malefyt. R.; Coffman, R.L.; O'Garra, A. Interleukin-10 and the interleukin-10 receptor. Annu Rev Immunol. 2001, 19, 683-765.

3. Saraiva, M.; O'Garra, A. The regulation of IL-10 production by immune cells. Nat Rev Immunol. 2010, 10(3), 170-81.

4. Kang, M.J.; Jang, A.R.; Park, J.Y.; Ahn, J.H.; Lee, T.S.; Kim, D.Y.; Lee, M.S.; Hwang, S.; Jeong, Y.J.; Park, J.H. IL-10 Protects Mice From the Lung Infection of Acinetobacter baumannii and Contributes to Bacterial Clearance by Regulating STAT3-Mediated MARCO Expression in Macrophages. Front Immunol. 2020, 11, 270.

5. Aziz, F.; Chakraborty, A.; Liu, K.; Zhang, T.; Li, X.; Du, R.; Monts, J.; Xu, G.; Li, Y.; Bai, R.; Dong, Z. Gastric tumorigenesis induced by combining Helicobacter pylori infection and chronic alcohol through IL-10 inhibition. Carcinogenesis. 2022, 43(2), 126-139.

6. Khatun, M.M.; Islam, M.A.; Baek, B.K. In Vitro and In Vivo IFN-γand IL-10 Measurement in Experimental Brucella abortus Biotype 1 Infection in Sprague-Dawley Rats. Vector Borne Zoonotic Dis. 2021, 21(8), 579-585.

7. Guo, Z.J.; Zhao, Y.; Zhang, Z.D.; Li, Y.M. Interleukin-10-Mediated Lymphopenia Caused by Acute Infection with Foot-and-Mouth Disease Virus in Mice. Viruses. 2021, 13(12), 2358.

please describe the technique of sacrificying of the animals. How long after natural death of animals the organs were collected?

“In the present research, we selected cervical dislocation method, which was the most widely used method of painless execution in mouse experiments. The tissue of the mice died for lethal infection or being killed humanely was collected immediately and then were used for determining the loads of bacteria and histopathological examination.” was added to Line 115

Please add the details of Ab used for IHC.

Line 126 was corrected to “The slices were sealed with 10% goat serum, IL-10 primary antibody (GB11534, 1:400, Wuhan Servicebio Biotechnology Co., LTD, wuhan, China) was added and incubated overnight at 4℃. HRP labeled Goat Anti-Rabbit IgG (GB23303, 1:200, Wuhan Servicebio Biotechnology Co., LTD, wuhan, China) was added and incubated at room temperature for 30 minutes.

The Reviewer would advice ELISA test for cytokines check.

Yes, your advice was reasonable and acceptable. However, considering COVID-19 epidemic was serious in Haikou these days, it was a pity that we were not able to go to the lab to do further experiments in ten days review response time. We hoped we could finish all these validation experiments in the future.

Fig. 2& 3 please replace the slides with more distinct. The visual effect esp. of Fig. 3 is poor.

Yes, Figure. 2 and Figure. 3 were replaced with higher resolution ratio.

Part 3.5 is too briefly explained.

Line 302 was corrected to “ Eight genes related to viral protein interaction with cytokine and cytokine receptor pathway and TNF signaling pathway were selected randomly in order to evaluate the accuracy and reliability of RNA-seq result. The results of qRT-PCR validation showed that the mRNA expression of IL10, MLKL, SDHD, CXCL2, CXCL13 were up-regulated while the mRNA expression of CXCL10, CCR9, IRF1 were down-regulated. All these results were consistent with RNA-seq data, which confirmed the reliability of RNA-seq data in this study (Figure 10).”

line 307-308- number of animals & restricted number for tests are not the basis to conclude about the animal model. What was the host infection pathway? 

Previous study has illustrated that mouse being highly sensitive and susceptible to P. multocida could serve as a useful model for pathogenesis studies and challenge studies [8]. What’s more, the basic protocol on how to establish A. baumannii infection mouse model by intraperitoneal injection was described in [9]. As gram-negative bacteria , there were many similarities between P. multocida and A. baumannii. In our previous study, the protocol on establishing P. multocida infection mouse model by intraperitoneal injection was exactly the same as what has described in [9], including previous preparation, experimental methods, sample collection and postoperative observation.

We have conducted 3 repeated experiments previously and we have confirmed that such a dose (1.5×107CFU/mL) could establish the mice infection model successfully. In this study, 12 mice were intraperitoneal injected of P. multocida while 12 mice were intraperitoneal injected of equal PBS. The mice intraperitoneal injected of P. multocida showed depressed spirit, insensitive to external stimulation, disorderly hair and trembling while the mice intraperitoneal injected of PBS showed no symptoms. HE results showed that the mice in challenge group had obvious lesion in spleen while the mice in control group showed no change in spleen. In conclusion, we thought P. multocida infection mouse model by intraperitoneal injection was successfully established.

8. Kharb,S.; Charan, S. Mouse model of haemorrhagic septicaemia: dissemination and multiplication of Pasteurella multocida B:2 in vital organs after intranasal and subcutaneous challenge in mice. Vet Res Commun. 2013, 37(1), 59-63.

9. Harris,G.; et al. Mouse Models of Acinetobacter baumannii Infection. Curr Protoc Microbiol. 2017, 46, 6G.3.1-6G.3.23.

In the results part there is a lack of histological details and post-mortem observations of infected mice.

According to the suggestion of reviewer, the part of HE staining description in Discussion was added to line 205 as following: HE results showed that the splenic tissue capsule of mice in challenge group was not smooth (figure 2 A). Plenty of lymphocyte and macrophage in red pulp and white pulp were necrotic and denaturated in challenge group. The cell body was swollen and cytoplasm is vacuolar while the nuclear was broken. The demarcation between red and white pulp was vague. The number and the volume of white pulp were reduced, the splenic artery was not obvious. Diffuse extramedullary hematopoietic cells could be seen in the red pulp (figure 2 B). As a comparison, in control group, there was no pathological injury in mouse spleen (figure 2 C, D).

No blood/serum tests done?

We have to admit that your advice was reasonable and acceptable as previous suggestion. However, considering COVID-19 epidemic was serious in Haikou these days, it was a pity that we were not able to go to the lab to do further experiments in a short time. We hoped we could finish all these experiments in the future.

How was IHC analyzed? Image -J?

No, IHC was analyzed by CaseViewer (version 2.4, 3DHISTECH Ltd., Budapest, Hungary).

Please add the slide with necrotic spleen cells.

Yes, Figure. 2 B showed that in red pulp and white pulp, plenty of immune cells were necrotic and denaturated. The nuclear was broken. We thought when the cells showed the status described above, they could be considered damage and tending to necrotic.

The Authors are making lot of speculations on the basis of sequencing analysis & in silico analysis.

Yes, indeed based on existing analysis and consulting extensive literature we made a lot of speculations which meant reasonable, and we hoped theses speculations could help illustrate the mighty mechanism of P. multocida infecting mice and provide the clues for the further study. We hoped that these validations would be done in the future.

Why Authors are mentioning viral proteins discussing bacterial infection?

Yes, in line 248 we mentioned viral protein interaction with cytokine and cytokine receptor based on KEGG database. In fact, according to KEGG, viral protein interaction with cytokine and cytokine receptor (map04061) was part of cytokine-cytokine receptor interaction (map04060). We mentioned viral protein interaction with cytokine and cytokine receptor just because its p value was less than 0.05 (0.0000006) while the p value of cytokine-cytokine receptor interaction was greater than 0.05 (0.0059) so we thought it was meaningless to mention cytokine-cytokine receptor interaction.

Reviewer 3 Report

Dear Authors,

your manuscript about the IL10 influence is interesting and well prepeared in the context of the graphic form. The pictures and figures are well done. The area of the research is also interesting, but in my opinion the article needs improvement. See the comments

Comments:

lines 57-58: Reword the sentence, add more details about  the cells which are able to produce IL-10, not just "lymphocyte"

lines 62-68: in my opinion this part is off topic - Pasteurella is not common cause of renal  fibrosis, as you mentioned it affects mainly the respiratory tract. If your aim was to highlight the IL-10caused tissue alleviation, use some more references please (f.e The Role of the Anti-Inflammatory Cytokine Interleukin-10 in Tissue Fibrosis, The Role of an IL-10/Hyaluronan Axis in Dermal Wound Healing) or move this part to the Discussion section.

line 85: Did you placed your P.multocida sequence in any database (f.e. GenBank)? if yes, please add the sequence number (or the number of homologous sequence from GB)

line 87: Use the full name of TSB and the full specificaion (it should be Qingdao Hope Bio-Technology Co., Ltd., Qingdao, China

line 88: the same situation is with NBS and TSB - complete the description  in all manuscript, write the names of the cities  capital letters etc

line 89: please give the brand name of the incubator. The laboratory equipment and machines should be described

line 97: do you mean SPF Kunming mice? I tried to find the source of the mice, Hainan Institute of Materia Medica Co. LTD (haikou, China) in internet, but there was no results - the most simmilar name was School of Chinise Materia Medica. Why I can't find the Institute?

line 101: in my opinion the word "adoption" is unproriate - shoud it be adaptation? On the other hand, adaptation should take more time than 3 days

lines 101-102 Were the groups equal? 9:9?

lines 102-104 Have you got the number of the Committee permission? Add it please

110-114 Reword the sentences, because it is unclear

114-117 Describe please what do you mean writing "the sample" - each spleen was divided on 3 parts? It is not big organ, but technically it could be done (I'm not discussing about the quality of the results). Or maybe you gave 3 spleens for HP, 3 for bacteriology and 3 for extraction? the 117 line do not dispell the doubts -  "The spleen tissue was collected immediately after the mice died and was fixed with 4% paraformaldehyde" - the fixing makes the organ useless for microbiological examination and RNA extraction (DNA could be extracted with specific kits)

Paragraphs 2.4 and 2.5 still leaves the doubt what "the sample" is

143-144 What kind of reagents were used for transcription? You're writing about instructions, but you do not give the name of the producer (or protocol)

I'm not able to check the paragraphs 2.6-2.7 because I'm not working with this technology

204-206 Was the HP examination result calculated statistically? You're writing that number of cells increased significantly, and the 2.9  paragraph (about statistical analysis) was described too brifly for me

Figure 2 description is unclear for readers. Are the pictures just from infected mice? You  should show the comparison of the control group and the infected group  (Similar situation in Fig 3)

Table 2 is unclear - it suggest that this group has just 5 mice, when in text you wrote that 5 mice died in 24h (and the rest was euthanized). Reword the text and the table clearly

line 226 - 229: how many mice were in the group? You wrote about 6 samples 3 infected and 3 uninfected?

The table 3 suggest 3 mice uninfected (C-control?) and 3 infected (E-experimental) - is it correct? It is a small group for conclusions

303-304: Pasteurellosis is not a huge problem zoonosis. The infections occures, but in most cases, the low immunocompetence of human must occur to get the infection from the animal. During the necropsy, after hurting a hand the abscesses may occur

347-349 - in Materials and Methods you give the sequences of starters for IL10, MLKL, SDHD, IRF1, CXCL10, CXCL13, CCR9, CXCL2 genes as those which you will investigate in qPCR. Figure 10 shows the results of analysis for those genes. I do not understand why are you writing about the expression level of LBP, CD14, LY96, TLR-4 genes in the examined and control group, when you did not examined it. My doubts might be connected with the fact that I'm not a specialist with genetic data analysis,  I hope that others reviewers will refer to those fragments.

In my opinion the english should be improved, but as I'm not a native speaker so this topic need to be consult with the specialist. For sure there are some spelling mistakes, and unproper forms of verbs in the text. The proofreading is needed.

Author Response

Reviewer3 

Comments and Suggestions for Authors

Dear Authors,

your manuscript about the IL10 influence is interesting and well prepeared in the context of the graphic form. The pictures and figures are well done. The area of the research is also interesting, but in my opinion the article needs improvement. See the comments

Comments:

lines 57-58: Reword the sentence, add more details about the cells which are able to produce IL-10, not just "lymphocyte"

The sentence in Line 59 was corrected to “IL10 is an anti-inflammatory cytokine produced primarily by leukocyte, including T cells, B cells, monocytes, macrophages and dendritic cells (DCs) [1,2,3]. 

1. Medzhitov,R.; Shevach, E.M.; Trinchieri, G.; Mellor, A.L.; Munn, D.H.; Gordon, S.; Libby, P.; Hansson, G.K.; Shortman, K.; Dong, C.; Gabrilovich, D.; Gabryšová, L.; Howes, A.; O'Garra, Highlights of 10 years of immunology in Nature Reviews Immunology. Nat Rev Immunol. 2011, 11(10), 693-702.

2. Moore,K.W.; de. Waal, Malefyt. R.; Coffman, R.L.; O'Garra, Interleukin-10 and the interleukin-10 receptor. Annu Rev Immunol. 2001, 19, 683-765.

3. Saraiva,M.; O'Garra, The regulation of IL-10 production by immune cells. Nat Rev Immunol. 2010, 10(3), 170-81.

lines 62-68: in my opinion this part is off topic - Pasteurella is not common cause of renal  fibrosis, as you mentioned it affects mainly the respiratory tract. If your aim was to highlight the IL-10caused tissue alleviation, use some more references please (f.e The Role of the Anti-Inflammatory Cytokine Interleukin-10 in Tissue Fibrosis, The Role of an IL-10/Hyaluronan Axis in Dermal Wound Healing) or move this part to the Discussion section.

We would like to thank you caused the suggestion was valuable to our study.

Line 59 was corrected to “IL-10 is an anti-inflammatory cytokine produced primarily by leukocyte, including T cells, B cells, monocytes, macrophages and dendritic cells (DCs)[1,2,3]. As an important immunomodulatory factor, IL-10 plays a crucial part in the negative regulation of immune response. In a study of acinetobacter baumannii infection mice model, compared with wild type(wt) mice, the mortality of IL10-/- knockout mice was increased, pathological damage aggravated and the lungs produced excess pro-inflammatory cytokines and chemokines [4]. In a mouse chronic gastritis model induced by helicobacter pylori, the result showed that IL-10 inhibited the occurrence of gastric cancer induced by helicobacter pylori infection combined with alcohol intake [5]. In another study, IL-10 levels in the serum were significantly elevated at 21 and 28 days after infection, and IL-10 production was significantly elevated in spleen cells at 21, 28, 35, and 42 days after infection in a brucella infection rat model, suggesting that cytokines secreted by Th2 cells such as IL-10 played an immune response to chronic infection [6]. An increase in IL-10 was also detected in a foot-and-mouth disease virus(fmdv) infection mouse model [7]. In conclusion, IL-10 is an important immunomodulator during microbial infection, however, there was evidence indicating IL-10 was also a double-edged sword that could both promote pathogen persistence and limit immune response [4]. The research on the detailed mechanism of IL-10 regulating host immune response is still not completely clarified.

1. Medzhitov, R.; Shevach, E.M.; Trinchieri, G.; Mellor, A.L.; Munn, D.H.; Gordon, S.; Libby, P.; Hansson, G.K.; Shortman, K.; Dong, C.; Gabrilovich, D.; Gabryšová, L.; Howes, A.; O'Garra, A. Highlights of 10 years of immunology in Nature Reviews Immunology. Nat Rev Immunol. 2011, 11(10), 693-702.

2. Moore, K.W.; de. Waal, Malefyt. R.; Coffman, R.L.; O'Garra, A. Interleukin-10 and the interleukin-10 receptor. Annu Rev Immunol. 2001, 19, 683-765.

3. Saraiva, M.; O'Garra, A. The regulation of IL-10 production by immune cells. Nat Rev Immunol. 2010, 10(3), 170-81.

4. Kang,M.J.;Jang, A.R.; Park, J.Y.; Ahn, J.H.; Lee, T.S.; Kim, D.Y.; Lee, M.S.; Hwang, S.; Jeong, Y.J.; Park, J.H. IL-10 Protects Mice From the Lung Infection of Acinetobacter baumannii and Contributes to Bacterial Clearance by Regulating STAT3-Mediated MARCO Expression in Macrophages. Front Immunol. 2020, 11, 270.

5. Aziz,F.;Chakraborty, A.; Liu, K.; Zhang, T.; Li, X.; Du, R.; Monts, J.; Xu, G.; Li, Y.; Bai, R.; Dong, Z. Gastric tumorigenesis induced by combining Helicobacter pylori infection and chronic alcohol through IL-10 inhibition. Carcinogenesis. 2022, 43(2), 126-139.

6. Khatun, M.M.; Islam, M.A.; Baek, B.K. In Vitro and In Vivo IFN-γ and IL-10 Measurement in Experimental Brucella abortus Biotype 1 Infection in Sprague-Dawley Rats. Vector Borne Zoonotic Dis. 2021, 21(8), 579-585.

7. Guo, Z.J.; Zhao, Y.; Zhang, Z.D.; Li, Y.M. Interleukin-10-Mediated Lymphopenia Caused by Acute Infection with Foot-and-Mouth Disease Virus in Mice. Viruses. 2021, 13(12), 2358.

line 85: Did you placed your P.multocida sequence in any database (f.e. GenBank)? if yes, please add the sequence number (or the number of homologous sequence from GB)

Yes, we have uploaded P. multocida sequence to NCBI.

Line 86 was corrected to “P. multocida HN02 strain (NCBI accession number: cp037865) was separated from goat lung tissue sample died of pneumonia in Liaoning province by our laboratory.”

line 87: Use the full name of TSB and the full specificaion (it should be Qingdao Hope Bio-Technology Co., Ltd., Qingdao, China

Yes, all “TSB” “TSA” “NBS” were corrected to “Tryptone Soy Broth (TSB)(Qingdao Haibo Biotechnology Co., LTD, Qingdao, China) ”, “Tryptone Soy Agar (TSA)(Qingdao Haibo Biotechnology Co., LTD, Qingdao, China) ”, “Newborn Bovine Serum (NBS)(Zhejiang Tianhang Biotechnology Co., LTD, Huzhou, China) ”.

line 88: the same situation is with NBS and TSB - complete the description in all manuscript, write the names of the cities capital letters etc

Yes, this problem has already been solved.

line 89: please give the brand name of the incubator. The laboratory equipment and machines should be described

Yes, “incubator” was corrected to “thermostatic oscillator (HZQ-X100A, Shanghai Yiheng Scientific Instrument Co., LTD, Shanghai, China)”.

line 97: do you mean SPF Kunming mice? I tried to find the source of the mice, Hainan Institute of Materia Medica Co. LTD (haikou, China) in internet, but there was no results - the most simmilar name was School of Chinise Materia Medica. Why I can't find the Institute?

Yes, in our study we used SPF Kunming mice bought from Hainan Institute of Medicine Co., Ltd. (Haikou, China).

line 101: in my opinion the word "adoption" is unproriate - shoud it be adaptation? On the other hand, adaptation should take more time than 3 days

Yes, we thought “adaptation” was more appropriate. Considering that raising the mice for too long might make them too heavy so that the challenge dose might be adjusted according to the weight.

lines 101-102 Were the groups equal? 9:9?

Line 104 was corrected to “The mice were randomly divided into a control group (n=12) and a challenge group (n=12) which was challenged with P. multocida. In challenge group, 5 mice were used for bacteria loads examination, 4 mice were used for HE staining examination and IHC while the other 3 mice were used for RNA-seq. In control group, 5 mice were used for bacteria loads examination, 4 mice were used for HE staining examination and IHC while the other 3 mice were used for RNA-seq.

lines 102-104 Have you got the number of the Committee permission? Add it please

Yes, Line 106 was corrected to “All experimental protocols were approved by the Academic Committee of  Hainan University under the ethical approval code HNUAUCC-2021-00068.”

110-114 Reword the sentences, because it is unclear

Line 113 was corrected to “All the mice were monitored for 7 days until they died. When mice appeared serious clinical symptoms (such as depression, appetite loss, shortness of breath, hair disorder) were considered to be dying, could be killed humanely. In the present research, we selected cervical dislocation method, which was the most widely used method of painless execution in mouse experiments. The spleen of the mice died for lethal infection or being killed humanely was collected immediately and then were used for determining the loads of bacteria and histopathological examination.”

114-117 Describe please what do you mean writing "the sample" - each spleen was divided on 3 parts? It is not big organ, but technically it could be done (I'm not discussing about the quality of the results). Or maybe you gave 3 spleens for HP, 3 for bacteriology and 3 for extraction? the 117 line do not dispell the doubts -  "The spleen tissue was collected immediately after the mice died and was fixed with 4% paraformaldehyde" - the fixing makes the organ useless for microbiological examination and RNA extraction (DNA could be extracted with specific kits)

The samples meant all the spleen tissues of every mouse used in our study. Considering the weight of each mouse spleen was less than 0.1g, we thought it was inappropriate to separate one single spleen into several parts. In fact we grouped the use of the mouse spleens: In challenge group, 5 mouse spleens were used for determining bacteria loads, 4 mouse spleens were used for HE staining and IHC while the other 3 mouse spleens were used for extracting RNA.

Line 117 was corrected to “The spleen samples of every mouse were divided into several parts: one part (challenge group: 3, control group: 3) was placed in liquid nitrogen for subsequent RNA and protein extraction, one part (challenge group: 5, control group: 5)  was used for bacteria loads determination, and one part (challenge group: 4, control group: 4) was soaked in paraformaldehyde for HE staining to observe pathological changes and IHC experiment.”

Paragraphs 2.4 and 2.5 still leaves the doubt what "the sample" is

Yes, we have already explained the doubt on the last question and corrected as following:The samples meant all the spleen tissues of every mouse used in our study and every spleen was divided into groups for different usage. 

143-144 What kind of reagents were used for transcription? You're writing about instructions, but you do not give the name of the producer (or protocol)

Line 147 was corrected to “The RNA without ribosome was randomly interrupted and re-transcribed into first strand and second strand cDNA following the instruction of reverse transcription kit (KR118, Tiangen Biotech (Beijing) Co., LTD, Beijing, China) after qualifying RNA examination.”

I'm not able to check the paragraphs 2.6-2.7 because I'm not working with this technology

Paragraph 2.6-2.7 were routine analysis for transcriptome sequencing article. For example, [7], [8] both had similar description.

In this study, mice spleen RNA was firstly extracted to construct a cDNA library and compared with the reference genome(GRCm38). Taking FPKM as an indicator to measure gene expression level. In this study, FPKM≧1 was considered to be meaningful. Then the genes of challenge group and control group were compared which process was called Differentially Expressed Genes (DEGs) analysis. DEGs were compared to GO and KEGG databases for functional annotation and classification. Finally, STRING was used for further protein-protein interaction analysis of DEGs of interest.

7. Wu, C.; Qin, X.; Li, P.; Pan, T.; Ren, W.; Li, N.; Peng, Y. Transcriptomic Analysis on Responses of Murine Lungs to Pasteurella multocida Infection. Front Cell Infect Microbiol. 2017, 7, 251.

8. Priya, G.B.; Nagaleekar, V.K.; Milton, A.A.P.; Saminathan, M.; Kumar, A.; Sahoo, A.R.; Wani, S.A.; Kumar, A.; Gupta, S.K. Genome wide host gene expression analysis in mice experimentally infected with Pasteurella multocida. PLoS One. 2017, 12(7), e0179420.

204-206 Was the HE examination result calculated statistically? You're writing that number of cells increased significantly, and the 2.9 paragraph (about statistical analysis) was described too brifly for me

We would like to appreciate for the valuable suggestion.

In our study HE examination was not calculated statistically. In fact, the HE examination result of challenge group was obviously different from control group so that we could find the increase or decrease in quantity of immune cells and the changes in cell morphology. In general, we focused on the the change of morphology of cells in spleen rather than counting the number of cells.

Line 195 was corrected to “All data was represented by mean ± standard error of means (SEM). GraphPadPrism8.0 was adopted for bacteria loads determination (Two-way ANOVA method) and qRT-PCR validation (Multiple t test). The nbinom test in DEseq2 was applied for the calculation of P value and FoldChange (FC). The difference was considered significant when P value<0.05.”

Figure 2 description is unclear for readers. Are the pictures just from infected mice? You should show the comparison of the control group and the infected group  (Similar situation in Fig 3)

Detailed legend was added to figure 2 and figure 3. Figure 2 A and B were different magnification figures of mice spleen in challenge group and figure 2 C and D were different magnification figures of mice spleen in control group. The legend of figure 2 and 3 were added as following:Figure 2. Results of HE staining. (A): 5×. Figure of spleens in challenge group. Compared with control group, the splenic tissue capsule of mice in challenge group was not smooth. (B): 20×. Figure of spleens in challenge group. Plenty of immune cells in red pulp and white pulp were necrotic and denaturated while nuclear was broken. The cell body was swollen and cytoplasm was vacuolar in challenge group. (C): 5×. Figure of spleens in control group. Compared with challenge group, the structure and morphology of spleen tissue in the control group were more intact. (D): 20×. Figure of spleens in control group. The boundary between red pulp and white pulp was distinct in control group.

Figure 3. Results of IHC. (A): 20×. Figure of spleens in challenge group. Compared with control group, IL-10 positive signal in mouse spleens in challenge group was increased significantly and was mostly expressed in lymphocytes and macrophages. (B): 20×. Figure of spleens in control group. Compared with challenge group, there was no obvious aggregation of IL-10 positive cells in control group.

Table 2 is unclear - it suggest that this group has just 5 mice, when in text you wrote that 5 mice died in 24h (and the rest was euthanized). Reword the text and the table clearly.

In our study we selected 5 mice died for lethal infection in challenge group and 5 mice in control group to determine bacteria loads.

Line 202 was corrected to “We selected 5 mice died for lethal infection in challenge group and 5 mice in control group to determine bacteria loads. The result of autopsy showed that the mouse spleen in challenge group turned to black, hyperemia and swelling (Figure 1 C) while in control group the mouse spleen showed no pathological changes (Figure 1 D). The result of bacterial loads showed that the spleens of 5 mice in challenge group all had plenty of bacterial colonization and the average of bacteria loads was 3.065×106 CFU/g  (Table 2).”

line 226 - 229: how many mice were in the group? You wrote about 6 samples 3 infected and 3 uninfected?

Yes, in the part of RNA-sequencing, we selected the rest of mice (3 mice in challenge group and 3 mice in control group). In challenge group, 5 spleens were used for bacteria loads determination, 4 spleens were used for HE staining and IHC experiment and 3 spleens were used for RNA extraction and RNA-seq. Similarly, in control group, 5 spleens were used for bacteria loads determination, 4 spleens were used for HE staining and IHC experiment and 3 spleens were used for RNA extraction and RNA-seq.

The table 3 suggest 3 mice uninfected (C-control?) and 3 infected (E-experimental) - is it correct? It is a small group for conclusions

Yes, “C” meant “control group” and “E” meant experiment group. We mentioned in the previous question that in the part of RNA-sequencing, we selected the rest of mice (3 mice in challenge group and 3 mice in control group). In deed it was a small group however it satisfied the requirement of minimum biological repetition so that the conclusion was reasonable.

303-304: Pasteurellosis is not a huge problem zoonosis. The infections occures, but in most cases, the low immunocompetence of human must occur to get the infection from the animal. During the necropsy, after hurting a hand the abscesses may occur

Yes, indeed zoonoses caused by pasteurella multocida were not so serious problems, but it could not be denied that zoonoses caused by pasteurella multocida were the problem that the public health industry could not ignore. One patient was reported to have developed arthritis after being scratched by a cat carrying pasteurella multocida [9]; Another cirrhosis patient developed pasteurella multocida bacteremia after living with animals for a long time [10]. In today's world, domestic pets are becoming more and more common. How to prevent and control zoonoses caused by pasteurella multocida is an urgent problem to be solved.

9. Shih, C.Y.; Chen, H.Y. Pasteurella multocida in total knee prosthetic joint infection caused by cat scratches and bites in a liver transplant recipient. IDCases. 2022, 29, e01560.

10. Rueda, Prada.L.; Cardozo, M.; Hudson, A.; McDermott, M.; Urbina, Verjel. D.C.; Dumic, I. Disseminated Pasteurella multocida in a patient with liver cirrhosis and spontaneous bacterial peritonitis - The role of cirrhosis-associated immune dysfunction. IDCases. 2022, 29, e01542.

347-349 - in Materials and Methods you give the sequences of starters for IL10, MLKL, SDHD, IRF1, CXCL10, CXCL13, CCR9, CXCL2 genes as those which you will investigate in qPCR. Figure 10 shows the results of analysis for those genes. I do not understand why are you writing about the expression level of LBP, CD14, LY96, TLR-4 genes in the examined and control group, when you did not examined it. My doubts might be connected with the fact that I'm not a specialist with genetic data analysis, I hope that others reviewers will refer to those fragments.

Yes, we truly selected these 8 immune-related genes for qRT-PCR randomly. In discussion we found some other interesting genes by consulting the literature which didn’t conflict with the genes mentioned previously. We might validate the expression level of LBP, CD14, LY96, TLR-4 in the future.

In my opinion the english should be improved, but as I'm not a native speaker so this topic need to be consult with the specialist. For sure there are some spelling mistakes, and unproper forms of verbs in the text. The proofreading is needed.

Yes, we have checked the overall manuscript carefully and corrected several minor errors.

Round 2

Reviewer 1 Report

The author's responses are convincing, however, it is not clearly input into the manuscript. The manuscript will be better if the rationale or importance can be more clearly described in the introduction.

Author Response

Reviewer1

Comments and Suggestions for Authors

The author's responses are convincing, however, it is not clearly input into the manuscript. The manuscript will be better if the rationale or importance can be more clearly described in the introduction. 

We really appreciated for your valuable suggestions, We didn’t input some responses into the  previous manuscript because we thought that was explanation for the question, might not appropriate to be added to the manuscript. According to your suggestion, combined with the replies to the review report (round 1), the paragraph now has been modified and added to the introduction part in Line 78 as following:

It is well known that the main difference between Gram-negative and positive bacteria is the various membrane structure [18], which is also one of the reasons for the different pathogenesis and symptoms of the host of bacterial infection. For example, as a gram-positive bacterium, Staphylococcus aureus was widespread and could cause severe diseases including bacteremia, pneumonia and cellulitis [19,20]; while P. multocida could cause infections in a wide range of animals and humans as a gram-negative bacteria [21]. Additionally, there were similarities and differences in the immune-related pathways involved in host interaction between Gram-positive and Gram-negative bacteria. For example, in order to explore the possible mechanism of the damage of bovine mammary gland caused by Staphylococcus aureus, bovine mammary fibroblasts were infected with Staphylococcus aureus, and the enriched DEGs involved in cytokine-cytokine receptor interaction, MAPK signaling pathway, TNF signaling pathway, among which IL-17 signaling pathway and Nod-like receptor signaling pathway were up-regulated [22]; in a study of mouse lungs infected with P. multocida, Toll-like receptor signaling pathway, cytokine-cytokine receptor interaction, NF-kB signaling pathway, Nod-like receptor signaling pathway were significantly up-regulated [23]. To sum up, even if different mechanisms of bacteria infection were explored, some of the enriched pathways might be the same, but the specific genes and regulatory mechanisms still remained discrepant. Therefore, as an important Gram-negative bacteria, the pathogenesis of P. multocida was of great necessity to explore. 

It is worth noting that the current studies have explored the pathogenic mechanism of P. multocida infection of the host target organ lung mostly. It is well known that as an important immune organ of the body, spleen is the main site for immune cells to swallow, recognize, and deliver antigens to induce immune reaction in vivo [24]. However, the related immune reaction, mechanism and the change of related gene expression were still unknown after P. multocida infecting animal organisms. RNA-seq technology has been applied to seek for the new insights into molecular mechanisms and the interaction relationship between hosts and pathogens, for example, in a P. multocida infection goat model, Zhang et al. [25] attempted to figure out the effects of P. multocida infection on goats and its specific pathogenic mechanism based on RNA-seq. Additionally, IHC was utilized in the molecular analysis of rabbits infected with P. multocida [26]. The application of the above technologies is increasing nowadays. In our present study, we established serotype A P. multocida infecting mouse spleen model. Then the immune reactions of mouse spleen to P. multocida infection were investigated by bacterial loads measurement, HE staining, IHC detection, RNA-seq and qRT-PCR, so as to provide basis for further study of the interaction between pathogen and host.   

18. Cao, Y.; Lin, H.C. Characterization and function of membrane vesicles in Gram-positive bacteria. Appl Microbiol Biotechnol. 2021, 105(5), 1795-1801.

19. Kobayashi, S.D.; Malachowa, N.; DeLeo, F.R. Pathogenesis of Staphylococcus aureus abscesses. Am J Pathol. 2015, 185(6), 1518-27.

20. Lowy, F.D. How Staphylococcus aureus adapts to its host. N Engl J Med. 2011, 364(21), 1987-90.

21. Peng, Z.; Wang, X.; Zhou, R.; Chen, H.; Wilson, B.A.; Wu, B. Pasteurella multocida: Genotypes and Genomics. Microbiol. Mol. Rev. 2019, 83, e00014-19.

22. Miao, Z.; et al. Transcriptome sequencing reveals fibrotic associated-genes involved in bovine mammary fibroblasts with Staphylococcus aureus. Int J Biochem Cell Biol. 2020, 121, 105696.

23. Wu, C.; Qin, X.; Li, P.; Pan, T.; Ren, W.; Li, N.; Peng, Y. Transcriptomic Analysis on Responses of Murine Lungs to Pasteurella multocida Infection. Front Cell Infect Microbiol. 2017, 7, 251.

24. The human spleen as the center of the blood defense system. Int J Hematol. 2020, 112(2), 147-158.

25. Zhang, W.; Jiao, Z.; Huang, H.; Wu, Y.; Wu, H.; Liu, Z.; Zhang, Z.; An, Q.; Cheng, Y.; Chen, S.; Man, C.; Du, L.; Wang, F.; Chen Q. Effects of Pasteurella multocida on Histopathology, miRNA and mRNA Expression Dynamics in Lung of Goats. Animals (Basel). 2022, 12(12), 1529.

26. Uenoyama, K.; Ueno, Y.; Tosaki, K.; Abeto, Y.; Ito, H.; Katsuda, K.; Shibahara, T. Immunohistochemical and molecular analysis of Pasteurella multocida in a rabbit with suppurative pleuropneumonia. J Vet Med Sci. 2020, 82(1), 89-93.

Reviewer 3 Report

Dear Authors,

thank you for article improvement according to reviewers suggestion. The Methods stayed unclear.

Lines 126-129 suggest using different mice for sampling:

"In challenge group, 5 mice were used for bacteria loads determination, 4 mice were used for HE staining examination and IHC while the other 3 mice were used for RNA-seq. In control group, 5 mice were used for 128
bacteria loads determination, 4 mice were used for HE staining examination and IHC while the other 3 mice were used for RNA-seq"

while lines 149-154 inform about dividing the spleens of each individual

"The spleen samples of every mouse were divided into several parts: one
part (challenge group: 3, control group: 3) was placed in liquid nitrogen for subsequent RNA and protein extraction, one part (challenge group: 5, control group: 5) was used for bacteria loads determination, and one part (challenge group: 4, control group: 4) was soaked in paraformaldehyde for HE staining to observe pathological changes and IHC experiment."

Please write the names  of cities with capital letters (it's written as ordinary word f.e shanghai)

Figures 2 and 3 have unclear description - you have 2x the same letter which is consterning...or maybe those pictureas are not deleted from the previous version? check it carefully please

Author Response

Reviewer3 

Comments and Suggestions for Authors

Dear Authors,

thank you for article improvement according to reviewers suggestion. The Methods stayed unclear.

Lines 126-129 suggest using different mice for sampling: "In challenge group, 5 mice were used for bacteria loads determination, 4 mice were used for HE staining examination and IHC while the other 3 mice were used for RNA-seq. In control group, 5 mice were used for bacteria loads determination, 4 mice were used for HE staining examination and IHC while the other 3 mice were used for RNA-seq."

while lines 149-154 inform about dividing the spleens of each individual: "The spleen samples of every mouse were divided into several parts: one part (challenge group: 3, control group: 3) was placed in liquid nitrogen for subsequent RNA and protein extraction, one part (challenge group: 5, control group: 5) was used for bacteria loads determination, and one part (challenge group: 4, control group: 4) was soaked in paraformaldehyde for HE staining to observe pathological changes and IHC experiment."

We really appreciated for your valuable suggestions and it was a pity that we didn’t make it clear in the previous manuscript. We modified it as following:

Line 126-129: The mice were randomly divided into two treatments, including a control group (n=12) and a challenge group (n=12) which was challenged with P. multocida. In each treatment, 5 mice were used for bacteria loads determination, 4 mice were used for HE staining examination and IHC while the other 3 mice were used for RNA-seq.

Lines 149-154: The spleen samples of each group were divided into three units: one unit (challenge group: 3, control group: 3) was placed in liquid nitrogen to extract RNA for subsequent RNA-seq, one unit (challenge group: 5, control group: 5) was used for bacteria loads determination, and one unit (challenge group: 4, control group: 4) was soaked in paraformaldehyde for HE staining to observe pathological changes and IHC experiment.

Please write the names of cities with capital letters (it's written as ordinary word f.e shanghai)

Yes, we have checked the manuscript and all the similar mistakes were corrected.

Figures 2 and 3 have unclear description - you have 2x the same letter which is consterning...or maybe those pictures are not deleted from the previous version? check it carefully please.

We appreciated for your patient examination. However after checking these two figures carefully, we didn’t find out the problems mentioned above. Maybe the version of the manuscript you read was different from the one we uploaded. We have uploaded the latest version of manuscript and we believed this could solve the doubts.